# EDEL: ERROR-DRIVEN ENSEMBLE LEARNING FOR IMBALANCED DATA CLASSIFICATION

## ABSTRACT

The class imbalance problem poses a critical challenge in high-stakes applications such as fraud detection, where the minority class often represents rare but consequential cases. In such settings, misclassifying minority instances can lead to substantial financial loss, underscoring the need for learning algorithms that remain reliable under severe imbalance. While deep learning methods have achieved remarkable success across various domains, their effectiveness often depends on large-scale datasets, and their black-box nature limits their interpretability, which is a critical requirement in high-stakes scenarios. To address this gap, we propose **Error-Driven Ensemble Learning** (EDEL), an adaptive machine learning algorithm that dynamically introduces misclassified instances during training, thereby placing greater emphasis on hard-to-classify samples. Through theoretical analysis and extensive experiments on multiple real-world datasets, EDEL demonstrates strong effectiveness, particularly under challenging imbalanced conditions.

## 1 INTRODUCTION

Real-world classification tasks are often affected by class imbalance, where the distribution of classes is highly skewed, as in fraud detection, credit risk assessment, and medical diagnosis. In these scenarios, the minority class typically corresponds to rare but critical instances (Gabidolla et al., 2024; Shahana et al., 2023). The inherent imbalance in such tasks biases machine learning models toward the majority class, leading them to overlook minority instances (Loffredo et al., 2024). This bias significantly degrades the model's performance on the minority class and undermines the reliability of decision-making in these systems (Sun et al., 2006). For instance, in fraud detection, fraudulent transactions occur infrequently but may cause substantial financial losses if overlooked, while in credit risk assessment and medical diagnosis, minority errors can result in severe financial or health-related consequences.

Traditional learning algorithms are inherently biased toward the majority class, yielding superficially high accuracy while neglecting minority samples. This results in degraded recall, a critical indicator of the reliability and trustworthiness of deployed models. To address this, researchers have proposed various techniques, including resampling methods (Maldonado et al., 2022; Sağlam & Cengiz, 2022; Abedin et al., 2022; Dixit & Mani, 2023; Yan et al., 2023) cost-sensitive learning (Elkan, 2001; Zhou & Liu, 2006; Ling & Sheng, 2010; Zhang & Hu, 2013; Cao et al., 2021), and advanced ensemble approaches(Freund & Schapire, 1997b; Chawla et al., 2003; Sağlam & Cengiz, 2022; Abedin et al., 2022; Zhao et al., 2025). Despite these advancements, accurately identifying minority instances remains a central challenge in imbalanced learning. In this work, we emphasize two key considerations: the need to effectively address *hard-to-classify* samples and the importance of ensuring interpretability in model decisions.

**C1. Requirement of attention on *hard-to-classify* samples.** A key concern in imbalanced learning is the presence of *hard-to-classify* samples that remain difficult to predict accurately, whether they belong to the minority or majority class. Such cases often arise from feature overlap, noise, or atypical patterns. Neglecting these samples undermines model robustness, as errors tend to concentrate on precisely those instances that are most informative for improving generalization. Figure 1 illustrates how class imbalance skews classification performance and error distribution. While the majority class is generally well recognized, a nontrivial portion of samples are misclassified as minority, placing erroneous signals into the minority space. These misclassified majorities increase

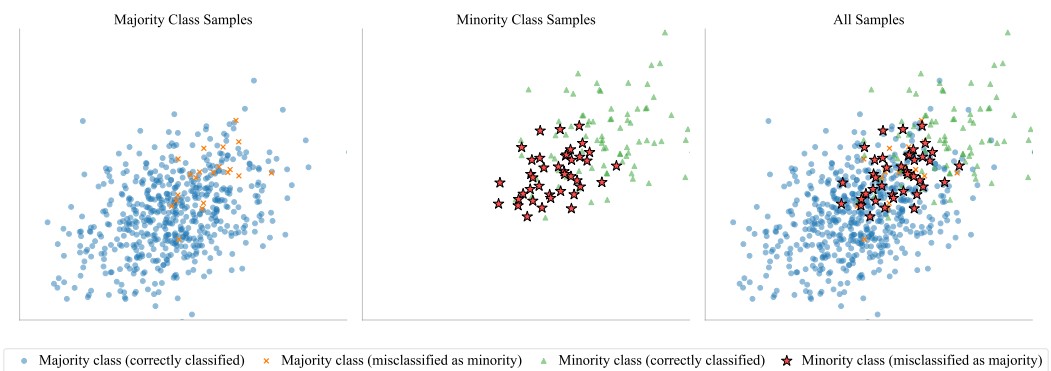

Figure 1: Impact of class imbalance on classification accuracy and error distribution.

overlap between classes, making the minority region appear less distinct, and further skew the decision boundary against the true minority. As a result, the classifier not only struggles with minority recognition but also inherits bias from misleading majority instances. We conjecture that these misclassifications occur because minority samples can closely resemble majority patterns, while some majority outliers deviate from typical distributions. Such concentration of errors is problematic: both minority instances misclassified as majority and majority outliers misclassified as minority contribute to the degradation of performance (especially the recall) and the erosion of applications' trustworthiness.

**C2. Requirement of attention to interpretability.** Another concern that should receive attention in imbalanced learning is interpretability, which is particularly critical in high-stakes real-world applications. Complex models, such as models based on Deep Learning algorithms, may achieve strong predictive performance but often operate as black boxes, making it difficult to understand why certain minority instances are misclassified or how decision boundaries are shaped by skewed data distributions. The lack of transparency hinders trust, complicates regulatory compliance, and prevents practitioners from diagnosing whether errors stem from data imbalance, noise, or just model bias. Therefore, beyond improving the predictive accuracy, methods for imbalanced classification should provide interpretable mechanisms for explaining decisions, especially for minority predictions where mistakes carry disproportionate costs, particularly in these real-world applications.

With these challenges in mind, we propose **E**rror-**D**riven **E**nsemble **L**earning (EDEL), a novel machine learning approach that progressively focuses on misclassified samples, a.k.a., *hard-to-classify* samples, into the learning process. By iteratively emphasizing them, EDEL refines its understanding and decision boundaries, improving recognition of the minority class while correcting misleading majority errors. Moreover, EDEL operates by partitioning the training data into multiple subsets, training weak classifiers, reinjecting misclassified samples, and aggregating predictions through probability averaging to ensure robust performance. This error-driven design naturally enhances interpretability by revealing where classifiers fail and how corrections are applied. Overall, the main contributions of EDEL are as follows:

- We highlight two critical concerns in imbalanced learning: the prevalence of *hard-to-classify* samples and the lack of interpretability.

- We introduce EDEL, a novel machine learning approach that progressively focuses on misclassified samples during training, significantly improving the performance on *hard-to-classify* samples. Experimental results on seven real-world datasets within imbalance ratios ranging from 1.54 to 577.88 demonstrate EDEL's superiority over baselines.

- We provide a theoretical foundation for EDEL, showing via McDiarmid's inequality and Bayes' theorem that dynamically incorporating *hard-to-classify* instances reduces empirical error, refines decision boundaries, and ensures asymptotic consistency with robust generalization under extreme imbalance. This theoretical guarantee contributes to the broader task of imbalanced classification by establishing a principled basis for designing reliable and interpretable learning algorithms.

## 2 PRELIMINARY

**Problem Definition**. For a $K$-class classification task, we have a training dataset with imbalanced distribution, consisting of $m$ samples: $\mathcal{D} = \{(\mathcal{X}, \mathcal{Y})\} = \{(\boldsymbol{x}_1, \boldsymbol{y}_1), \ldots, (\boldsymbol{x}_m, \boldsymbol{y}_m)\}$, where $\boldsymbol{x}_i \in \mathbb{R}^d$ represents the $d$-dimensional features and $\boldsymbol{y}_i$ is the corresponding ground truth. Without loss of generality, we assume the class sample sizes are sorted in descending order, i.e., $n_1 \geq n_2 \geq \cdots \geq n_K$, where $n_k$ denotes the number of samples in class $k$. Our objective is to conduct a scheme $\pi(\mathcal{X}; \theta)$ that accurately infers the label for $\mathcal{X}$ under the class-balanced setting, i.e., $\boldsymbol{\pi}(\mathcal{X}; \theta) \mapsto \mathcal{Y}$.

Within this setup, we further introduce the *hard-to-classify* and *easy-to-classify* samples in this work. Let $(\boldsymbol{x}, \boldsymbol{y})$ be an instance, let $\mathcal{F} = \{f_1, \cdots, f_{|\mathcal{F}|}\}$ denote a set of classifiers.

**DEFINITION 1:** *Easy-to-classify* samples correspond to data points that are well-aligned with their class distribution and consistently captured by different classifiers. That means,

$$f_i(\boldsymbol{x}) = \boldsymbol{y}, \quad \forall f_i \in \mathcal{F}. \tag{1}$$

**DEFINITION 2:** *Hard-to-classify* samples lie in regions of feature overlap, noise, or atypical structure, where even a subset of classifiers fail. In this work, an instance $(\boldsymbol{x}, \boldsymbol{y})$ is *Hard-to-classify* if it is misclassified by at least one classifier in $\mathcal{F}$, namely,

$$\exists f_i \in \mathcal{F} \quad \text{such that} f_i(\boldsymbol{x}) \neq \boldsymbol{y}. \tag{2}$$

## 3 METHODOLOGIES

### 3.1 PIPELINE

Initially, we randomly form $N$-subsets from original dataset, i.e., $\mathcal{D} \rightarrow \{\mathcal{D}_i\}_{i=1}^N$, where $\sum_{i=1}^N |\mathcal{D}_i| = |\mathcal{D}|$. Herein, subset $\mathcal{D}_i$ can be viewed as a partial observation drawn from the true data distribution. By partitioning the data into multiple subsets, EDEL generates diverse local views of the distribution, while each subset has its opportunity to emphasize certain characteristics or patterns. Whereafter, weak classifiers $\hat{\boldsymbol{\pi}}$ are conducted on corresponding subset as,

$$\hat{\boldsymbol{y}} = \hat{\boldsymbol{\pi}}_i(\boldsymbol{x}), \ \boldsymbol{x} \in \mathcal{D}_i, \ i \in [1, N], \tag{3}$$

and *hard-to-classify* samples can be formed via these weak classifiers $\hat{\boldsymbol{\pi}}$ by,

$$\mathcal{D}_i^h \leftarrow \{(\boldsymbol{x}, \boldsymbol{y}) \mid \hat{\boldsymbol{\pi}}_i(\boldsymbol{x}) \neq \boldsymbol{y}, \ (\boldsymbol{x}, \boldsymbol{y}) \in \mathcal{D} \backslash \mathcal{D}_i\}. \tag{4}$$

Therefore, we update $\mathcal{D}_i$ by incorporating $\mathcal{D}_i^h$ as,

$$\mathcal{D}_i \leftarrow \mathcal{D}_i \cup \mathcal{D}_i^h. \tag{5}$$

Notably, we have $\sum_{i=1}^N |\mathcal{D}_i| > |\mathcal{D}|$ after this operation.

Once the weak classifiers are well-trained, EDEL integrates parameters from each $\hat{\boldsymbol{\pi}}(\cdot)$ to form the final classifier $\boldsymbol{\pi}(\cdot)$ by employing a probability-averaging scheme, namely,

$$\Theta = \frac{1}{N} \sum_{i=1}^N \hat{\Theta}_i, \quad \text{where} \quad \hat{\Theta}_i \leftarrow \hat{\boldsymbol{\pi}}_i(\mathcal{D}_i). \tag{6}$$

Algorithm 1 outlines the training process of the proposed EDEL. To ensure generality, we denote the training cost of a weak classifier as $T_{\boldsymbol{\pi}}$, and divide EDEL's overall complexity into 5 stages:

(1) **Data Partitioning.** Using stratified sampling to preserve class distribution on a dataset $\mathcal{D}$ of size $m$, this step has time complexity $\mathcal{O}(m)$.

(2) **Weak Classifier Training.** Training $N$ weak classifiers $\hat{\boldsymbol{\pi}}$ on each subset $\mathcal{D}_i$ cost, giving a total complexity of $\mathcal{O}(N \times T_{\boldsymbol{\pi}})$.

(3) **Subset Update.** Each $\boldsymbol{\pi}_i$ conducts prediction on remaining $N-1$ subsets, for a total of $\mathcal{O}((N-1) \times m)$.

---

**Algorithm 1:** Training Process of EDEL.

**Input:** $\mathcal{D}, N, \hat{\boldsymbol{\pi}}$

1 Initially partition $\mathcal{D} \rightarrow \{\mathcal{D}_i\}_{i=1}^N$
2 **while** *not done* **do**
3     **for** $i = 1$ *to* $N$ **do**
4         **for** $(\boldsymbol{x}, \boldsymbol{y}) \in \mathcal{D}_i$ **do**
5             $\hat{\boldsymbol{y}} = \hat{\pi}_i(\boldsymbol{x})$
6         **end**
7         $\Theta_i \leftarrow \hat{\boldsymbol{\pi}}_i(\mathcal{D}_i)$
8         $\mathcal{D}_i^h \leftarrow \{\emptyset\}$
9         **for** $(\boldsymbol{x}, \boldsymbol{y}) \in \mathcal{D} \backslash \mathcal{D}_i$ **do**
10             $\hat{\boldsymbol{y}} = \hat{\pi}_i(\boldsymbol{x})$
11             **if** $\hat{\boldsymbol{y}} \neq \boldsymbol{y}$ **then**
12                 $\mathcal{D}_i^h \leftarrow \mathcal{D}_i^h \cup \{(\boldsymbol{x}, \boldsymbol{y})\}$
13                 $(\mathcal{D} \backslash \mathcal{D}_i) \backslash \{(\boldsymbol{x}, \boldsymbol{y})\}$
14             **end**
15         **end**
16         $\mathcal{D}_i \leftarrow \mathcal{D}_i \cup \mathcal{D}_i^h$
17     **end**
18 **end**
19 $\Theta \leftarrow \frac{1}{N} \sum_{i=1}^N \hat{\Theta}_i$

**Output:** $\pi_\Theta$

---

    (4) **Training Cycle**. Retraining each $\hat{\boldsymbol{\pi}}_i$ on updated subset $\mathcal{D}i$ costs $T_{\boldsymbol{\pi}}$, totaling $\mathcal{O}(N \times T\boldsymbol{\pi})$.

    (5) **Ensemble Strategy:** Forming the final classifier by averaging outputs costs $\mathcal{O}(N)$ per sample, totaling $\mathcal{O}(N \times m)$.

Thus, the overall time complexity can be summed up by the 5 stages as:

$$\mathcal{O}(m) + \mathcal{O}(N \times T_{\boldsymbol{\pi}}) + \mathcal{O}((N-1) \times m) + \mathcal{O}(N \times T_{\boldsymbol{\pi}}) + \mathcal{O}(N \times m) = \mathcal{O}(N \times (T_{\boldsymbol{\pi}} + m)).$$

In summary, the time complexity of the EDEL algorithm is linear with respect to both the weak classifier training time $T_{\boldsymbol{\pi}}$ and the dataset size $m$, and it scales proportionally with the number of weak classifiers $N$. Through data partitioning and parallel computation, EDEL effectively scales to large datasets. In highly imbalanced scenarios, adjusting the number of classifiers allows EDEL to balance high performance with computational efficiency, making it a versatile solution for large-scale and imbalanced data.

## 3.2 THEORETICAL FOUNDATION

**Data Aspect.** EDEL employs stratified sampling to split training dataset $\mathcal{D}$ into $N$ non-overlapping subsets $\{\mathcal{D}_i\}_{i=1}^N$ initially. Following the definition on Sec. 2, the class proportion for class $k$ is given by $p_k = \frac{n_k}{m}$. Stratified sampling allocates $\frac{n_k}{N}$ samples of class $k$ to each subset $\mathcal{D}_i$, resulting in a subset size of $\frac{m}{N}$ (assuming $m$ is divisible by $N$ for simplicity). The proportion of class $k$ in $\mathcal{D}_i$ can be described as,

$$p_{i,k} = \frac{\frac{n_k}{N}}{\frac{m}{N}} = \frac{n_k}{m} = p_k. \tag{7}$$

Ideally, $p_{i,k} = p_k$, ensuring that each subset $\mathcal{D}_i$ mirrors the overall class distribution of $\mathcal{D}$. However, due to finite sampling, random allocation may introduce deviations. The number of class $k$ samples $n_{i,k}$ in $\mathcal{D}_i$ follows a hypergeometric distribution, with expectation $\mathbb{E}[n_{i,k}] = \frac{n_k}{N}$ and variance:

$$\text{Var}(n_{i,k}) = \frac{n_k}{N} \left(1 - \frac{1}{N}\right) \frac{m - n_k}{m - 1}. \tag{8}$$

The deviation of the proportion is defined as $\Delta p_{i,k} = |p_{i,k} - p_k|$. Using the Chebyshev inequality, the probability that this deviation exceeds $\delta$ is bounded by:

$$P(|\Delta p_{i,k}| \geq \delta) \leq \frac{\text{Var}(n_{i,k})}{n^2 \delta^2}, \tag{9}$$

where $n = \frac{m}{N}$. As the total sample size $m \to \infty$, the variance term $\frac{\text{Var}(n_{i,k})}{n^2} \to 0$, implying that $\Delta p_{i,k} \to 0$. This confirms that $p_{i,k} \approx p_k$ with high probability for large $m$, thus establishing the representativeness of $\mathcal{D}_i$ with respect to $\mathcal{D}$. This representativeness supports the dynamic enhancement of minority class samples, as detailed in Appendix B.

**Classifier Aspect.** Weak classifiers $\hat{\boldsymbol{\pi}}_i^{(t+1)}$ are trained on the dynamically enhanced subset $\mathcal{D}_i^{(t)} = \mathcal{D}_i^{(t-1)} \cup \mathcal{D}_i^{(t-1),h}$, with its training error rate:

$$\hat{\epsilon}_i^{(t+1)} = \frac{1}{|\mathcal{D}_i^{(t)}|} \sum_{(\boldsymbol{x},y) \in \mathcal{D}_i^{(t)}} \mathbb{I}\big(\hat{\boldsymbol{\pi}}_i^{(t+1)}(\boldsymbol{x}) \neq \boldsymbol{y}\big), \tag{10}$$

where $\mathbb{I}(\cdot)$ is the indicator function.

Assume $\mathcal{D}_i^{(t)}$ contains $n_i^{(t)}$ samples, with $n_{i,0}^{(t)}$ samples from the majority class (class 0) and $n_{i,1}^{(t)}$ samples from the minority class (class 1), such that $n_i^{(t)} = n_{i,0}^{(t)} + n_{i,1}^{(t)}$. For a simple weak classifier, such as a decision tree stump (which performs a single split), the optimal classification rule minimizes the training error rate by selecting a feature and threshold that partitions the data into two regions, assigning the majority class label to each region. In the degenerate case without any split (equivalent to a constant classifier predicting the majority class), the minimum training error rate is the minority class proportion:

$$\hat{\epsilon}_i^{(t+1)} = \frac{n_{i,1}^{(t)}}{n_i^{(t)}}. \tag{11}$$

However, with splitting enabled, the actual training error rate is strictly lower than this baseline, as the split allows the classifier to capture more nuanced patterns in the data. Dynamic enhancement increases the number of minority class samples $n_{i,1}^{(t)}$ (due to the higher proportion of minority class instances in $\mathcal{D}_i^h$), making the subset $\mathcal{D}_i^{(t)}$ more balanced. This balance enables the classifier to identify better decision boundaries, reducing the actual $\hat{\epsilon}_i^{(t+1)}$ compared to the previous iteration or to random guessing (where $\epsilon = 0.5$ for a balanced dataset). Since $\rho' > \rho$ (where $\rho$ and $\rho'$ are the minority class proportions in $\mathcal{D}_i^{(t-1)}$ and $\mathcal{D}_i^{(t)}$, respectively), $n_{i,1}^{(t)}$ grows relative to $n_{i,0}^{(t)}$, ensuring the actual $\epsilon < 0.5$ and improving performance on minority class instances. This minority class enrichment is rigorously derived using Bayes' theorem in Appendix B, with further analysis of extreme imbalance scenarios.

To assess the generalization performance of weak classifiers within EDEL, we employ Rademacher complexity (Bartlett & Mendelson, 2003) $\mathcal{R}_{n_i^{(t)}}(\mathcal{H})$, which measures the capacity of the hypothesis space $\mathcal{H}$ (e.g., decision stumps) over the training set $\mathcal{D}_i^{(t)}$. It is defined as:

$$\hat{\mathcal{R}}_{n_i^{(t)}}(\mathcal{H}) = \mathbb{E}_\sigma \left[ \sup_{h \in \mathcal{H}} \left| \frac{2}{n_i^{(t)}} \sum_{j=1}^{n_i^{(t)}} \sigma_j h(\boldsymbol{x}_j) \right| \ \middle| \ \mathcal{D}_i^{(t)} \right], \tag{12}$$

where $\sigma_j \in \{-1, +1\}$ are independent Rademacher random variables, and the expectation is taken over their distribution conditional on the $\mathcal{D}_i^{(t)}$. The unconditional Rademacher complexity is $\mathcal{R}_{n_i^{(t)}}(\mathcal{H}) = \mathbb{E}[\hat{\mathcal{R}}_{n_i^{(t)}}(\mathcal{H})]$. For a decision tree with $d$ features, $\mathcal{R}_{n_i^{(t)}}(\mathcal{H}) = \mathcal{O}(\sqrt{d/n_i^{(t)}})$, based on the empirical VC-dimension bound (where the VC-dimension of decision stumps is $O(d)$). Thereby, the true generalization error $\epsilon_i^{(t+1)}$ (i.e., the expected error over the data distribution) is bounded using Rademacher complexity, with probability at least $1 - \delta$:

$$\epsilon_i^{(t+1)} \leq \hat{\epsilon}_i^{(t+1)} + \frac{\mathcal{R}_{n_i^{(t)}}(\mathcal{H})}{2} + \sqrt{\frac{\ln(1/\delta)}{2n_i^{(t)}}}, \tag{13}$$

where $\delta > 0$ is a confidence parameter. As $n_i^{(t)}$ increases, $\mathcal{R}_{n_i^{(t)}}(\mathcal{H}) = \mathcal{O}(\sqrt{d/n_i^{(t)}})$ and $\sqrt{\frac{\ln(1/\delta)}{2n_i^{(t)}}}$ both decrease, ensuring that the empirical error $\hat{\epsilon}_i^{(t+1)}$ is a good approximation of the true error $\epsilon_i^{(t+1)}$. This confirms that the weak classifier generalizes well from the enhanced training subset $\mathcal{D}_i^{(t)}$.

**Convergence Aspect.** Building on the stratified sampling and dynamic enhancement mechanisms outlined in the **Data Aspect** and **Classifier Aspect**, we analyze the convergence of the empirical error $\hat{\epsilon}_i^{(t+1)}$ to the true error $\epsilon_i^{(t+1)}$ for the proposed EDEL. The empirical error is defined in Equation equation 10 as:

$$\hat{\epsilon}_i^{(t+1)} = \frac{1}{|\mathcal{D}_i^{(t)}|} \sum_{(\boldsymbol{x},\boldsymbol{y}) \in \mathcal{D}_i^{(t)}} \mathcal{Z}_i(\boldsymbol{x},\boldsymbol{y}), \tag{14}$$

where $\mathcal{Z}_i(\boldsymbol{x},\boldsymbol{y}) = \mathbb{I}(\hat{\boldsymbol{\pi}}_i^{(t+1)}(\boldsymbol{x}) \neq \boldsymbol{y})$ is the indicator function for misclassification, and $\epsilon_i^{(t+1)} = \mathbb{E}_{(\boldsymbol{x},\boldsymbol{y}) \sim \mathcal{D}}[\mathcal{Z}_i(\boldsymbol{x},\boldsymbol{y})]$ is the true error under the data distribution $\mathcal{D}$. The initial stratified sampling ensures approximate i.i.d. conditions (Equation equation 7), with class proportion deviations $\Delta p_{i,k} \to 0$ as $m \to \infty$ (Sec. 3.2). However, the dynamic incorporation of misclassified samples $\mathcal{D}_i^h$ into $\mathcal{D}_i^{(t)}$ introduces dependencies and shifts the subset distribution toward minority class instances (Appendix B).

Under approximate i.i.d. conditions, the expected empirical error is approximately unbiased:

$$\mathbb{E}[\hat{\epsilon}_i^{(t+1)}] = \mathbb{E}\left[\frac{1}{|\mathcal{D}_i^{(t)}|} \sum_{(\boldsymbol{x},\boldsymbol{y}) \in \mathcal{D}_i^{(t)}} \mathcal{Z}_i(\boldsymbol{x},\boldsymbol{y})\right] \approx \frac{1}{|\mathcal{D}_i^{(t)}|} \sum_{(\boldsymbol{x},\boldsymbol{y}) \in \mathcal{D}_i^{(t)}} \mathbb{E}[\mathcal{Z}_i(\boldsymbol{x},\boldsymbol{y})] \approx \epsilon_i^{(t+1)}, \tag{15}$$

The linearity of expectation ensures the first equality holds regardless of independence, but potential distribution shifts from dynamic enhancement may cause $\mathbb{E}[\mathcal{Z}_i(\boldsymbol{x},\boldsymbol{y})]$ to vary across samples, introducing a bias $\left|\mathbb{E}[\hat{\epsilon}_i^{(t+1)}] - \epsilon_i^{(t+1)}\right| \leq \mathcal{O}(\sqrt{d/|\mathcal{D}_i^{(t)}|})$, as controlled by the Rademacher complexity of the loss function (Bartlett & Mendelson, 2003).

To address dependencies introduced by the enhancement process, we employ the McDiarmid's inequality (McDiarmid, 1989), which requires only bounded differences rather than strict independence. Consider $\hat{\epsilon}_i^{(t+1)}$ as a function of the training samples in $\mathcal{D}_i^{(t)}$. Changing one sample $(\boldsymbol{x}_j, \boldsymbol{y}_j)$ to $(\boldsymbol{x}_j', \boldsymbol{y}_j')$ affects $\mathcal{Z}_j$, altering $\hat{\epsilon}_i^{(t+1)}$ by at most $\frac{1}{|\mathcal{D}_i^{(t)}|}$, as $\mathcal{Z}_j \in [0,1]$. Thus, the bounded difference constant is $c_j = \frac{1}{|\mathcal{D}_i^{(t)}|}$, and the sum of squared differences is $\sum_{j=1}^{|\mathcal{D}_i^{(t)}|} c_j^2 = \frac{1}{|\mathcal{D}_i^{(t)}|}$. The McDiarmid's inequality yields:

$$P\left(\left|\hat{\epsilon}_i^{(t+1)} - \mathbb{E}[\hat{\epsilon}_i^{(t+1)}]\right| \geq \delta\right) \leq 2\exp\left(-2\delta^2 |\mathcal{D}_i^{(t)}|\right). \tag{16}$$

This ensures that $\hat{\epsilon}_i^{(t+1)}$ concentrates around its expectation with high probability as the subset size $|\mathcal{D}_i^{(t)}|$ increases.

To bridge $\mathbb{E}[\hat{\epsilon}_i^{(t+1)}]$ to $\epsilon_i^{(t+1)}$, we leverage the Rademacher complexity bound (Equation equation 13) (Bartlett & Mendelson, 2003). For the hypothesis space $\mathcal{H}$ of weak classifiers (e.g., decision stumps), the true error is bounded with probability at least $1 - \delta$:

$$\epsilon_i^{(t+1)} \leq \hat{\epsilon}_i^{(t+1)} + \frac{\mathcal{R}_{|\mathcal{D}_i^{(t)}|}(\mathcal{H})}{2} + \sqrt{\frac{\ln(1/\delta)}{2|\mathcal{D}_i^{(t)}|}}, \tag{17}$$

where $\mathcal{R}_{|\mathcal{D}_i^{(t)}|}(\mathcal{H}) = \mathcal{O}(\sqrt{d/|\mathcal{D}_i^{(t)}|})$ for decision stumps with $d$ features. The bias $\left|\mathbb{E}[\hat{\epsilon}_i^{(t+1)}] - \epsilon_i^{(t+1)}\right|$ is thus bounded by $\mathcal{O}(\sqrt{d/|\mathcal{D}_i^{(t)}|})$, mitigating the impact of non-identical distributions caused by dynamic enhancement.

Combining the McDiarmid's inequality and Rademacher bounds, we establish that $\hat{\epsilon}_i^{(t+1)}$ converges to $\epsilon_i^{(t+1)}$ with high probability, at least $1 - 2\exp(-2\delta^2 |\mathcal{D}_i^{(t)}|) - \delta$, where $|\mathcal{D}_i^{(t)}| = |\mathcal{D}_i^{(t-1)}| + |\mathcal{D}_i^h|$

reflects the growth from dynamic enhancement. This convergence is robust to the dependencies and distribution shifts introduced by $(\mathcal{D}_i^h$ and the ensemble averaging further mitigates subset-specific biases, ensuring robust performance. Further analysis of minority class enrichment and integrated classifier performance is provided in Appendix B and Appendix B.3.

## 4 EXPERIMENTS

**Dataset.** We conduct experiments on seven publicly available, real-world datasets that span diverse domains, detailed in Table 1. They exhibit a wide range of imbalance ratios (IR), from moderate (e.g., SBD with IR=1.54) to extreme (e.g., CCFD with IR=577.88), making them ideal for evaluating the robustness of our method across varying degrees of class imbalance. Each dataset has been preprocessed to handle missing values and eliminate redundant features. More details of datasets and metrics are described in Appendix C.1.

Table 1: Summary of the datasets used in the experiments.

| Dataset | Samples | Pos | Neg | Features | IR |
|---|---|---|---|---|---|
| **SBD** | 4,601 | 1,813 | 2,788 | 57 | 1.54 |
| **AID** | 48,842 | 11,687 | 37,155 | 13 | 3.18 |
| **TCD** | 30,000 | 6,636 | 23,364 | 23 | 3.52 |
| **CDH** | 253,680 | 35,346 | 218,334 | 21 | 6.18 |
| **BMD** | 41,188 | 4,640 | 36,548 | 20 | 7.88 |
| **GMSC** | 150,000 | 10,026 | 139,974 | 10 | 13.96 |
| **CCFD** | 284,807 | 492 | 284,315 | 29 | 577.88 |

**Baseline.** We evaluate EDEL against five baselines, including (1) basic method, the Original instances with no imbalance handling applied; (2) data-level methods, SMOTE (Chawla et al., 2002), RandomUnderSampler (RUS) (He & Garcia, 2009) and SMOTE-TLNN-DEPSO (S-T-D) (Dixit & Mani, 2023); (3) ensemble learning methods, MESA (Liu et al., 2020) and CHRE (Zhao et al., 2025). All methods, including EDEL and the baselines, are evaluated using four classifiers: Decision Tree (DT), Random Forest (RF), XGBoost (XGB), and LightGBM (LGBM). Experimental results are reported by 5-fold stratified cross-validation, with each fold allocating 80% for training and 20% for testing. More details of classifier parameters are described in Appendix C.2.

Table 2: Results in terms of AUC.

| Clf | Mth | SBD | AID | TCD | CDH | BMD | GMSC | CCFD |
|---|---|---|---|---|---|---|---|---|
| DT | Orig | 0.9030 ± 0.0075 | 0.7717 ± 0.0041 | 0.6143 ± 0.0074 | 0.5979 ± 0.0031 | 0.7294 ± 0.0066 | 0.6120 ± 0.0047 | 0.8921 ± 0.0041 |
| | RUS | 0.9062 ± 0.0139 | 0.7824 ± 0.0036 | 0.6214 ± 0.0041 | 0.6514 ± 0.0031 | 0.8367 ± 0.0044 | 0.7017 ± 0.0033 | 0.9012 ± 0.0137 |
| | SMOTE | 0.9126 ± 0.0112 | 0.7780 ± 0.0060 | 0.6131 ± 0.0073 | 0.5979 ± 0.0054 | 0.7451 ± 0.0085 | 0.6358 ± 0.0055 | 0.8953 ± 0.0164 |
| | MESA | 0.9655 ± 0.0134 | 0.8721 ± 0.0052 | 0.7366 ± 0.0074 | 0.7734 ± 0.0056 | 0.9322 ± 0.0036 | 0.8145 ± 0.0128 | 0.9609 ± 0.0177 |
| | S-T-D | 0.8893 ± 0.0085 | 0.7765 ± 0.0030 | 0.6165 ± 0.0073 | 0.6210 ± 0.0046 | 0.7935 ± 0.0098 | 0.6648 ± 0.0096 | 0.9085 ± 0.0180 |
| | CHRE | 0.8993 ± 0.0128 | 0.7840 ± 0.0025 | 0.6383 ± 0.0104 | 0.6633 ± 0.0052 | 0.8226 ± 0.0038 | 0.6932 ± 0.0038 | 0.9123 ± 0.0054 |
| | EDEL | **0.9911 ± 0.0182** | **0.9519 ± 0.0724** | **0.9187 ± 0.1536** | **0.9241 ± 0.1381** | **0.9745 ± 0.0475** | **0.9457 ± 0.1107** | **0.9827 ± 0.0387** |
| RF | Orig | 0.9860 ± 0.0041 | 0.8954 ± 0.0036 | 0.7639 ± 0.0046 | 0.7974 ± 0.0025 | 0.9444 ± 0.0024 | 0.8387 ± 0.0055 | 0.9497 ± 0.0108 |
| | RUS | 0.9859 ± 0.0044 | 0.8973 ± 0.0020 | 0.7684 ± 0.0063 | 0.8077 ± 0.0023 | 0.9430 ± 0.0027 | 0.8519 ± 0.0038 | 0.9777 ± 0.0077 |
| | SMOTE | 0.9857 ± 0.0040 | 0.8902 ± 0.0027 | 0.7506 ± 0.0039 | 0.7942 ± 0.0032 | 0.9415 ± 0.0027 | 0.8207 ± 0.0055 | 0.9691 ± 0.0103 |
| | MESA | 0.9858 ± 0.0040 | 0.9015 ± 0.0044 | 0.7729 ± 0.0048 | 0.8168 ± 0.0024 | 0.9464 ± 0.0022 | 0.8450 ± 0.0052 | 0.9829 ± 0.0074 |
| | S-T-D | 0.9750 ± 0.0072 | 0.8942 ± 0.0025 | 0.7530 ± 0.0072 | 0.7970 ± 0.0029 | 0.9428 ± 0.0017 | 0.8381 ± 0.0041 | 0.9793 ± 0.0108 |
| | CHRE | 0.9428 ± 0.0074 | 0.8207 ± 0.0021 | 0.7112 ± 0.0089 | 0.7139 ± 0.0055 | 0.8782 ± 0.0045 | 0.7283 ± 0.0066 | 0.9104 ± 0.0107 |
| | EDEL | **0.9972 ± 0.0058** | **0.9656 ± 0.0448** | **0.9461 ± 0.1117** | **0.9506 ± 0.0841** | **0.9795 ± 0.0306** | **0.9583 ± 0.0641** | **0.9923 ± 0.0172** |
| XGB | Orig | 0.9872 ± 0.0031 | 0.9285 ± 0.0025 | 0.7661 ± 0.0059 | 0.8269 ± 0.0025 | 0.9458 ± 0.0018 | 0.8573 ± 0.0043 | 0.9789 ± 0.0062 |
| | RUS | 0.9869 ± 0.0030 | 0.9258 ± 0.0030 | 0.7579 ± 0.0073 | 0.8223 ± 0.0024 | 0.9411 ± 0.0030 | 0.8515 ± 0.0034 | 0.9780 ± 0.0086 |
| | SMOTE | 0.9871 ± 0.0027 | 0.9239 ± 0.0030 | 0.7378 ± 0.0084 | 0.8262 ± 0.0029 | 0.9398 ± 0.0025 | 0.8174 ± 0.0034 | 0.9763 ± 0.0110 |
| | MESA | 0.9872 ± 0.0024 | 0.9273 ± 0.0031 | 0.7715 ± 0.0043 | 0.8259 ± 0.0032 | 0.9463 ± 0.0018 | 0.8566 ± 0.0053 | 0.9727 ± 0.0132 |
| | S-T-D | 0.9782 ± 0.0047 | 0.9213 ± 0.0024 | 0.7370 ± 0.0084 | 0.8196 ± 0.0032 | 0.9431 ± 0.0027 | 0.8360 ± 0.0048 | 0.9784 ± 0.0077 |
| | CHRE | 0.9454 ± 0.0064 | 0.8235 ± 0.0037 | 0.6829 ± 0.0094 | 0.7402 ± 0.0034 | 0.8873 ± 0.0042 | 0.7704 ± 0.0075 | 0.9196 ± 0.0153 |
| | EDEL | **0.9968 ± 0.0065** | **0.9424 ± 0.0127** | **0.9039 ± 0.0897** | **0.8365 ± 0.0125** | **0.9708 ± 0.0244** | **0.8963 ± 0.0270** | **0.9968 ± 0.0071** |
| LGBM | Orig | 0.9882 ± 0.0035 | 0.9287 ± 0.0029 | 0.7805 ± 0.0052 | 0.8302 ± 0.0030 | 0.9508 ± 0.0019 | 0.8649 ± 0.0039 | 0.7549 ± 0.0368 |
| | RUS | 0.9880 ± 0.0029 | 0.9278 ± 0.0029 | 0.7765 ± 0.0051 | 0.8290 ± 0.0030 | 0.9456 ± 0.0029 | 0.8623 ± 0.0035 | 0.9812 ± 0.0104 |
| | SMOTE | 0.9881 ± 0.0032 | 0.9250 ± 0.0030 | 0.7544 ± 0.0083 | 0.8266 ± 0.0031 | 0.9437 ± 0.0031 | 0.8313 ± 0.0038 | 0.9682 ± 0.0166 |
| | MESA | 0.9880 ± 0.0028 | 0.9281 ± 0.0030 | 0.7818 ± 0.0066 | 0.8301 ± 0.0030 | 0.9496 ± 0.0022 | 0.8634 ± 0.0049 | 0.9682 ± 0.0143 |
| | S-T-D | 0.9796 ± 0.0050 | 0.9229 ± 0.0029 | 0.7522 ± 0.0105 | 0.8258 ± 0.0034 | 0.9459 ± 0.0026 | 0.8455 ± 0.0040 | 0.9745 ± 0.0109 |
| | CHRE | 0.9536 ± 0.0046 | 0.8209 ± 0.0066 | 0.6748 ± 0.0068 | 0.7483 ± 0.0038 | 0.8914 ± 0.0055 | 0.7749 ± 0.0101 | 0.9152 ± 0.0158 |
| | EDEL | **0.9969 ± 0.0065** | **0.9366 ± 0.0079** | **0.8429 ± 0.0474** | **0.8352 ± 0.0024** | **0.9679 ± 0.0152** | **0.8726 ± 0.0121** | **0.9965 ± 0.0079** |

**Performance.** Tables 2 and 3 report the results in terms of AUC and F1-measure. The best gain is highlighted in **bold** while the second best is underlined. Generally, across all settings, EDEL demon-

Table 3: Results in terms of F1-measure.

| Clf | Mth | SBD | AID | TCD | CDH | BMD | GMSC | CCFD |
|---|---|---|---|---|---|---|---|---|
| DT | Orig | 0.8819 ± 0.0088 | 0.6171 ± 0.0043 | 0.3999 ± 0.0104 | 0.3072 ± 0.0047 | 0.5155 ± 0.0119 | 0.2679 ± 0.0082 | 0.7741 ± 0.0272 |
| | RUS | 0.8834 ± 0.0171 | 0.6173 ± 0.0056 | 0.4208 ± 0.0043 | 0.3436 ± 0.0023 | 0.5402 ± 0.0071 | 0.2386 ± 0.0028 | 0.0312 ± 0.0043 |
| | SMOTE | 0.8914 ± 0.0140 | 0.6242 ± 0.0061 | 0.4048 ± 0.0097 | 0.3074 ± 0.0088 | 0.5233 ± 0.0151 | 0.2482 ± 0.0062 | 0.5405 ± 0.0289 |
| | MESA | 0.9047 ± 0.0262 | 0.6516 ± 0.0099 | 0.4949 ± 0.0099 | 0.4029 ± 0.0034 | 0.6109 ± 0.0078 | 0.3237 ± 0.0111 | 0.5625 ± 0.0375 |
| | S-T-D | 0.8673 ± 0.0102 | 0.6451 ± 0.0030 | 0.4070 ± 0.0100 | 0.3373 ± 0.0065 | 0.5686 ± 0.0125 | 0.2780 ± 0.0095 | 0.3996 ± 0.0295 |
| | CHRE | 0.8712 ± 0.0152 | 0.6292 ± 0.0028 | 0.4384 ± 0.0110 | 0.3650 ± 0.0043 | 0.5761 ± 0.0075 | 0.3140 ± 0.0051 | 0.7419 ± 0.0210 |
| | EDEL | **0.9709 ± 0.0412** | **0.8215 ± 0.1339** | **0.8096 ± 0.2554** | **0.7584 ± 0.2778** | **0.8375 ± 0.2023** | **0.7690 ± 0.2987** | **0.9182 ± 0.1289** |
| RF | Orig | 0.9406 ± 0.0095 | 0.6669 ± 0.0045 | 0.4711 ± 0.0086 | 0.2561 ± 0.0046 | 0.5781 ± 0.0086 | 0.2796 ± 0.0085 | 0.8588 ± 0.0186 |
| | RUS | 0.9425 ± 0.0078 | 0.6725 ± 0.0028 | 0.5212 ± 0.0073 | 0.4249 ± 0.0013 | 0.5934 ± 0.0065 | 0.3241 ± 0.0026 | 0.1157 ± 0.0190 |
| | SMOTE | 0.9415 ± 0.0055 | 0.6736 ± 0.0040 | 0.4959 ± 0.0092 | 0.2733 ± 0.0050 | 0.6120 ± 0.0139 | 0.3524 ± 0.0059 | 0.8565 ± 0.0272 |
| | MESA | 0.9380 ± 0.0060 | 0.6843 ± 0.0095 | 0.5353 ± 0.0118 | 0.4156 ± 0.0115 | 0.6440 ± 0.0074 | 0.3629 ± 0.0160 | 0.8243 ± 0.0311 |
| | S-T-D | 0.9189 ± 0.0101 | 0.6864 ± 0.0025 | 0.4886 ± 0.0116 | 0.3775 ± 0.0035 | 0.6381 ± 0.0076 | 0.3883 ± 0.0072 | 0.8405 ± 0.0273 |
| | CHRE | 0.9244 ± 0.0084 | 0.6720 ± 0.0029 | 0.5361 ± 0.0130 | 0.4501 ± 0.0032 | 0.6379 ± 0.0116 | 0.4296 ± 0.0112 | 0.8414 ± 0.0344 |
| | EDEL | **0.9824 ± 0.0209** | **0.8192 ± 0.1143** | **0.7448 ± 0.2229** | **0.6829 ± 0.1925** | **0.7145 ± 0.1194** | **0.5929 ± 0.1588** | **0.9370 ± 0.0780** |
| XGB | Orig | 0.9379 ± 0.0094 | 0.7122 ± 0.0066 | 0.4673 ± 0.0157 | 0.2610 ± 0.0030 | 0.5904 ± 0.0159 | 0.2884 ± 0.0128 | 0.8749 ± 0.0298 |
| | RUS | 0.9396 ± 0.0080 | 0.7056 ± 0.0043 | 0.5047 ± 0.0096 | 0.4362 ± 0.0025 | 0.5944 ± 0.0051 | 0.3217 ± 0.0030 | 0.0798 ± 0.0090 |
| | SMOTE | 0.9388 ± 0.0112 | 0.7158 ± 0.0052 | 0.4747 ± 0.0090 | 0.2832 ± 0.0041 | 0.6112 ± 0.0123 | 0.3448 ± 0.0026 | 0.8364 ± 0.0267 |
| | MESA | 0.9428 ± 0.0063 | 0.7201 ± 0.0082 | 0.5194 ± 0.0118 | 0.4378 ± 0.0150 | 0.6420 ± 0.0078 | 0.3273 ± 0.0293 | 0.8145 ± 0.0488 |
| | S-T-D | 0.9197 ± 0.0089 | 0.7169 ± 0.0037 | 0.4605 ± 0.0117 | 0.3686 ± 0.0053 | 0.6323 ± 0.0097 | 0.3874 ± 0.0045 | 0.7803 ± 0.0286 |
| | CHRE | 0.9293 ± 0.0103 | 0.6524 ± 0.0047 | 0.4819 ± 0.0093 | 0.4079 ± 0.0039 | 0.6020 ± 0.0073 | 0.3661 ± 0.0053 | 0.8295 ± 0.0275 |
| | EDEL | **0.9796 ± 0.0282** | **0.7378 ± 0.0316** | **0.6855 ± 0.1595** | **0.4508 ± 0.0296** | **0.7333 ± 0.1409** | **0.4821 ± 0.0827** | **0.9446 ± 0.0628** |
| LGBM | Orig | 0.9447 ± 0.0081 | 0.7124 ± 0.0049 | 0.4770 ± 0.0088 | 0.2506 ± 0.0029 | 0.6074 ± 0.0096 | 0.2895 ± 0.0138 | 0.3834 ± 0.1507 |
| | RUS | 0.9417 ± 0.0062 | 0.7110 ± 0.0049 | 0.5265 ± 0.0117 | **0.4412 ± 0.0026** | 0.5990 ± 0.0058 | 0.3323 ± 0.0060 | 0.0897 ± 0.0116 |
| | SMOTE | 0.9441 ± 0.0064 | 0.7162 ± 0.0053 | 0.5031 ± 0.0077 | 0.2977 ± 0.0035 | 0.6322 ± 0.0131 | 0.3613 ± 0.0015 | 0.7310 ± 0.0321 |
| | MESA | 0.9427 ± 0.0057 | 0.7211 ± 0.0086 | 0.5236 ± 0.0116 | 0.4371 ± 0.0096 | 0.6420 ± 0.0290 | 0.3515 ± 0.0214 | 0.8360 ± 0.0396 |
| | S-T-D | 0.9212 ± 0.0113 | 0.7206 ± 0.0052 | 0.4789 ± 0.0091 | 0.3820 ± 0.0051 | 0.6391 ± 0.0038 | **0.4019 ± 0.0048** | 0.7117 ± 0.0342 |
| | CHRE | 0.9390 ± 0.0055 | 0.6477 ± 0.0102 | 0.4720 ± 0.0064 | 0.4340 ± 0.0076 | 0.6028 ± 0.0132 | 0.3646 ± 0.0071 | 0.1877 ± 0.0964 |
| | EDEL | **0.9799 ± 0.0280** | **0.7282 ± 0.0201** | **0.5627 ± 0.0769** | 0.4039 ± 0.0181 | **0.6904 ± 0.1018** | 0.3988 ± 0.0228 | **0.9366 ± 0.0753** |

strates consistent and often substantial improvements in both AUC and F1-measure, highlighting its robustness in imbalanced scenarios. Specifically, we have the following observations.

(O1). On low to moderate imbalance datasets such as SBD (IR=1.54) and AID (IR=3.18), EDEL achieves near-perfect performance with RF that AUC up to 0.9972, F1=0.9824 on SBD, and consistently outperforms all baselines across both RF and DT classifiers.

(O2). For medium imbalance datasets including TCD (IR=3.52) and CDH (IR=6.18), EDEL delivers significant gains. Notably, on CDH, EDEL achieves an AUC of 0.9506 and 0.6829 F1 score with RF, substantially higher than Orig (AUC=0.7974, F1=0.2561).

(O3). On high imbalance datasets, BMD (IR=7.88) and GMSC (IR=13.96), EDEL exhibits strong resilience. For instance, on GMSC, DT with EDEL achieves 0.7690 F1 score compared to 0.2679 with the basic method (orig), evidencing its ability to recover minority information effectively.

(O4). The CCFD, with IR=577.88, is an extremely imbalanced scenario. EDEL remains robust and yields the superior performance on both AUC and F1 score across all classifier-dataset combinations.

Overall, the above observations confirm that EDEL consistently performs superiorly across varying imbalance ratios and classifier backbones, with notable improvements in terms of F1, validating its effectiveness in capturing minority-class signals without sacrificing overall discrimination power.

**Theoretical Consistency Validation.** The experimental results further validate the theoretical foundation of EDEL in addressing imbalanced classification tasks. On the extremely imbalanced CCFD dataset (IR=577.88), EDEL's error-driven update mechanism exhibits strong adaptability by dynamically re-injecting misclassified instances, which enables the model to continuously refine its decision boundary and improve minority class recognition as the number of weak classifiers increases. This progressive refinement directly reflects the convergence properties established in Section 3, where McDiarmid's inequality guarantees iterative error reduction with high probability. In practice, we observe that EDEL steadily improves performance up to a certain ensemble size, after which the gains diminish. For example, the marginal improvements between 4 and 5 weak classifiers confirm the theoretical prediction of diminishing returns, as the empirical error approaches its asymptotic bound and convergence stabilizes (details in Appendix B.3 and Appendix D.1). Importantly, these observations not only highlight EDEL's robustness on highly skewed data distributions but also demonstrate its alignment with the theoretical analysis, reinforcing confidence in its broad applicability to real-world imbalanced learning problems.

**Feature Distribution Analysis.** We present Table 4 to analyze the distribution shifts of features V1–V10 in GMSC, i.e., features change in $\mathcal{D}_i^{(t-1)}$) and ($\mathcal{D}_i^{(t)}$. The stratified sampling in Section 3.2

Table 4: Feature distribution shifts in EDEL's training subsets by class (GMSC, 5-fold CV).

| Feature | $\mathcal{D}_i^{(t-1),1}$ Mean | $\mathcal{D}_i^{t,1}$ Mean | % Change (minority) | $\mathcal{D}_i^{(t-1),0}$ Mean | $\mathcal{D}_i^{t,0}$ Mean | % Change (majority) | KS Statistic (p-value) |
|---|---|---|---|---|---|---|---|
| V1 | 4.37 | 3.82 | -12.4 | 6.17 | 7.42 | 20.2 | 0.106 ($< 10^{-236}$) |
| V2 | 45.93 | 46.34 | 0.9 | 52.75 | 51.65 | -2.1 | 0.049 ($< 10^{-46}$) |
| V3 | 2.39 | 1.81 | -24.3 | 0.28 | 0.48 | 70.8 | 0.075 ($< 10^{-120}$) |
| V4 | 295.12 | 296.13 | 0.3 | 357.15 | 359.26 | 0.6 | 0.017 ($< 10^{-5}$) |
| V5 | 5525.81 | 5621.75 | 1.7 | 6397.43 | 6362.33 | -0.5 | 0.028 ($< 10^{-13}$) |
| V6 | 7.88 | 8.11 | 2.9 | 8.49 | 8.49 | 0.0 | 0.020 ($< 10^{-8}$) |
| V7 | 2.09 | 1.48 | -29.4 | 0.14 | 0.29 | 116.8 | 0.052 ($< 10^{-53}$) |
| V8 | 0.99 | 1.01 | 2.6 | 1.02 | 1.03 | 0.8 | 0.020 ($< 10^{-8}$) |
| V9 | 1.83 | 1.30 | -29.1 | 0.13 | 0.26 | 107.0 | 0.041 ($< 10^{-36}$) |
| V10 | 0.93 | 0.91 | -2.1 | 0.72 | 0.77 | 6.1 | 0.022 ($< 10^{-7}$) |

Table 5: Analysis of five representative samples from the GMSC dataset.

| ID | label | DT | RF | XGB | LGB | EDEL | V1 | V2 | V3 | V4 | V5 | V6 | V7 | V8 | V9 | V10 | Type |
|---|---|---|---|---|---|---|---|---|---|---|---|---|---|---|---|---|---|
| 12057 | 0 | 0 | 0 | 0 | 0 | 0 | 0.0018 | 77 | 0 | 0.0000 | 5000 | 4 | 0 | 0 | 0 | 0 | Easy |
| 26462 | 1 | 1 | 1 | 1 | 1 | 1 | 0.9797 | 43 | 2 | 0.5431 | 2700 | 4 | 1 | 2 | 3 | 0 | Easy |
| 29011 | 1 | 0 | 0 | 0 | 0 | 1 | 0.3842 | 49 | 1 | 0.7734 | 1760 | 8 | 0 | 1 | 0 | 4 | Hard |
| 68219 | 1 | 0 | 0 | 0 | 0 | 1 | 0.9910 | 23 | 0 | 0.2045 | 1500 | 2 | 0 | 0 | 0 | 0 | Hard |
| 35849 | 0 | 1 | 1 | 1 | 1 | 0 | 1.0180 | 46 | 4 | 0.4021 | 5500 | 6 | 0 | 1 | 1 | 0 | Hard |

ensures representative subsets ($p_{i,k} \approx p_k$). For minority, dynamic enhancement reduces V1 (-12.4%, from high to moderate utilization), V3 (-24.3%, fewer 30–59 day overdue), V7 (-29.4%, fewer 90+ day overdue), and V9 (-29.1%, fewer 60–89 day overdue), while increasing V6 (+2.9%, more open loans), enhancing minority class signals. For majority, V1 increases (+20.2%, higher utilization), V3 (+70.8%), and V7 (+116.8%) increase, reflecting inclusion of hard-to-classify majority samples. The average KS statistic (0.043, all $p < 10^{-5}$) confirms significant distribution shifts, validating the Bayes' theorem prediction of minority class enrichment in $\mathcal{D}_i^h$. (See Appendix B.1).

**Case Study**. To investigate the impact of class imbalance on classification performance and to further validate the interpretability of EDEL, we provide case-level analyses in Table 5, covering both *easy-to-classify* and *hard-to-classify* instances from GMSC (IR=13.96). The corresponding feature details (v1-v10) are shown in Table 6, which lists each variable, its description, and its preferences with respect to delinquency risk. A downward arrow (↓) indicates that lower values are preferable, an upward arrow (↑) indicates that higher values are preferable, and a dash (–) denotes neutral influence. These financial indicators, such as delinquency counts (V3, V7, V9), credit utilization (V1), and income level (V5), are critical for understanding classification challenges in imbalanced settings. The results in Table 5 illustrate the capacity of EDEL on addressing *hard-to-classify* samples. For ambiguous minority cases (29011, 68219), baselines misclassify them as majority due to weak or absent minority-indicative features (lower v3, v7, v9). In contrast, EDEL leverages the reinjection of misclassified instances (a.k.a $\mathcal{D}^h$), allowing minority cases (35849) that other models misclassify as minority. Overall, EDEL corrects almost 75.39% of baseline minority hard errors, yielding a $2.87\times$ improvement in F1 (0.7690 vs. 0.2679, cf. Table 3), aligning with our theoretical foundations.

Table 6: Feature descriptions for the GMSC dataset.

| Feature ID | Feature | Description | Preference |
|---|---|---|---|
| – | SeriousDlqin2yrs (label) | Borrower experienced 90+ days past due delinquency | – |
| V1 | RevolvingUtilizationOfUnsecuredLines | Balance on credit cards and personal lines of credit (excluding real estate and installment debt) divided by credit limits | ↓ |
| V2 | Age | Borrower's age in years | – |
| V3 | NumberOfTime30-59DaysPastDueNotWorse | Times borrower was 30–59 days past due in the last 2 years | ↓ |
| V4 | DebtRatio | Monthly debt payments, alimony, and living costs divided by monthly gross income | ↓ |
| V5 | MonthlyIncome | Monthly income | ↑ |
| V6 | NumberOfOpenCreditLinesAndLoans | Number of open loans and lines of credit | – |
| V7 | NumberOfTimes90DaysLate | Times borrower was 90+ days past due | ↓ |
| V8 | NumberRealEstateLoansOrLines | Number of mortgage and real estate loans | – |
| V9 | NumberOfTime60-89DaysPastDueNotWorse | Times borrower was 60–89 days past due in the last 2 years | ↓ |
| V10 | NumberOfDependents | Number of dependents, excluding the borrower | ↓ |

**Visualization** Figure 2 visualizes the original training set $\mathcal{D}$ with corresponding five subsets produced after one EDEL update under stratified sampling (size=1000) on GMSC. In subfigure (a), positive samples are sparse and dispersed and heavily intermixed with dense negative clusters, echoing the imbalance-induced boundary bias we discussed in Section 1. In contrast, subfigures (b)–(f)

show clear minority enrichment: positives become denser and more coherent, and boundary overlap is reduced, which matches our theoretical analysis in Section 3 and Appendix B.1. Moreover, the five subsets illustrate subtle distributional differences, reflecting the multi-view partitioning strategy and contributing to ensemble diversity, further enabling inspection of iterative refinements on *hard-to-classify* samples to foster trust in high-stakes applications.

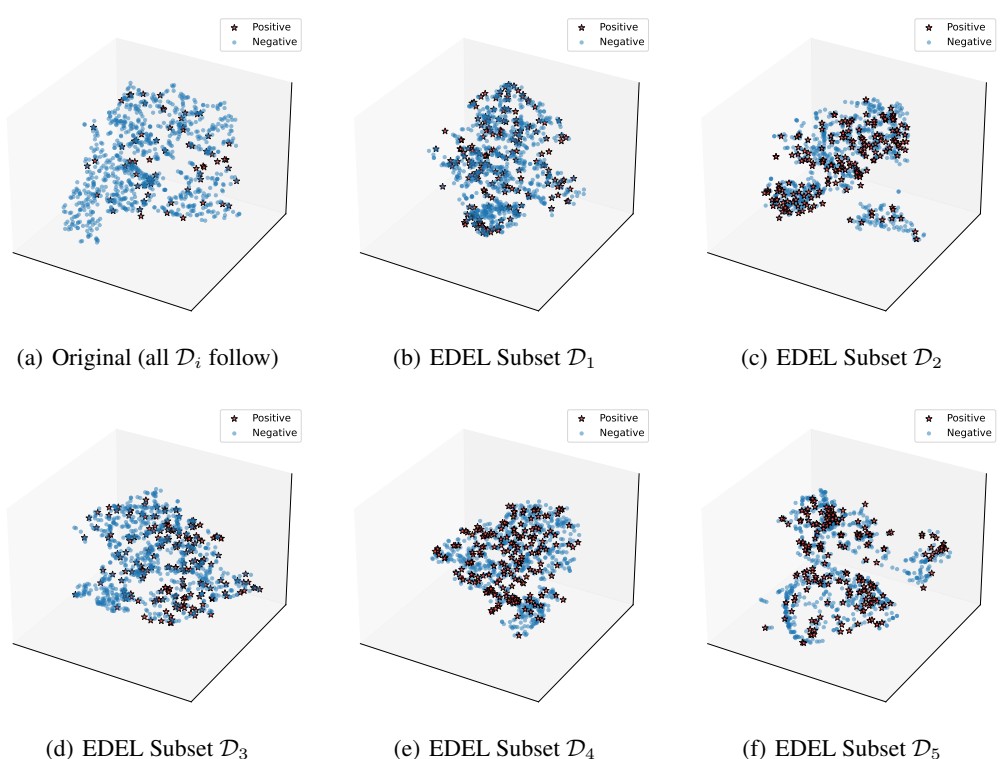

| (a) Original (all $\mathcal{D}_i$ follow) | (b) EDEL Subset $\mathcal{D}_1$ | (c) EDEL Subset $\mathcal{D}_2$ |

| (d) EDEL Subset $\mathcal{D}_3$ | (e) EDEL Subset $\mathcal{D}_4$ | (f) EDEL Subset $\mathcal{D}_5$ |

Figure 2: Visualization of data reinjection in EDEL.

Due to space limitation, additional discussions and related works are listed in Appendix D and E.

## 5 CONCLUSION

This work introduces EDEL, a novel algorithm developed to address the challenge of class imbalance in machine learning. By dynamically adjusting the learning process using misclassified samples and incorporating a multi-perspective learning approach, EDEL enhances the model's ability to recognize minority class instances. Extensive experiments on seven real-world datasets, including domains such as financial fraud detection and credit risk assessment, demonstrate that EDEL significantly improves key metrics, such as AUC and F1-measure, particularly under extreme class imbalance. These findings validate the robustness and adaptability of EDEL in managing both moderate and severe imbalances. Moreover, EDEL's adaptive nature makes it effective across a range of classifiers, consistently improving performance.

However, while EDEL significantly improves minority class recognition, particularly in high-risk domains, it introduces higher computational complexity, posing challenges for large-scale datasets. To address this, parallel and distributed computing methods could be considered. Overall, EDEL provides a robust and adaptable solution to the class imbalance problem, with considerable potential for future enhancements and broader real-world applications, particularly in fields where minority class recognition is critical.

## REPRODUCIBILITY STATEMENT

To ensure the reproducibility of our results, we provide the complete implementation of the EDEL algorithm and all experimental details. The source code is publicly available at `https://anonymous.4open.science/r/EDEL-1A3E/Readme.md`, including data preprocessing pipelines, algorithm codes, and experimental scripts. All experiments are conducted on seven publicly available benchmark datasets, with detailed URLs provided in Appendix C. We use 5-fold stratified cross-validation with fixed random seeds 42 and report detailed hyperparameters for all baseline methods and classifiers.

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

APPENDIX

## A    THE USE OF LARGE LANGUAGE MODELS (LLMS)

In the preparation of this manuscript, large language models (LLMs) were utilized to assist with language polishing and grammar checking. Specifically, an LLM was employed to refine the text for improved clarity, coherence, and grammatical accuracy. This ensured that the final version of the article maintains high linguistic standards while preserving the original ideas and content generated by the authors. The use of LLMs was limited to these supportive tasks, and no substantial content was generated by the models themselves.

## B    THEORETICAL ANALYSIS

This section extends the data representativeness and classifier performance analysis in Section 3.2, providing detailed derivations for minority class enrichment using Bayes' theorem.

**Misclassified Sample Aspect**. Weak classifiers trained on imbalanced data are prone to exhibiting biases toward the majority class, resulting in a higher misclassification rate for minority class samples. To quantify the proportion of minority class samples within $\mathcal{D}_i^h$ (cf Eq.( 4)), we derive the conditional probability using Bayes' theorem:

$$P(\boldsymbol{y} = 1 \mid (\boldsymbol{x}, \boldsymbol{y}) \in \mathcal{D}_i^h) = \frac{P((\boldsymbol{x}, \boldsymbol{y}) \in \mathcal{D}_i^h \mid \boldsymbol{y} = 1)P(\boldsymbol{y} = 1)}{P((\boldsymbol{x}, \boldsymbol{y}) \in \mathcal{D}_i^h)}. \tag{18}$$

Herein, the numerator represents misclassification samples, i.e., $\epsilon_i = P(\boldsymbol{\pi}_i(\boldsymbol{x}) \neq \boldsymbol{y})$. Using the law of total probability, $\epsilon_i$ can be expressed as:

$$\epsilon_i = P(\boldsymbol{\pi}_i(\boldsymbol{x}) \neq \boldsymbol{y} \mid \boldsymbol{y} = 0)P(\boldsymbol{y} = 0) + P(\boldsymbol{\pi}_i(\boldsymbol{x}) \neq \boldsymbol{y} \mid \boldsymbol{y} = 1)P(\boldsymbol{y} = 1), \tag{19}$$

where $P(y = 0) = 1 - \rho$ and $P(y = 1) = \rho$ (with $\rho \ll 0.5$ indicating class imbalance). Substituting the misclassification rates $\epsilon_{i,0} = P(\boldsymbol{\pi}_i(\boldsymbol{x}) \neq 0 \mid y = 0)$ and $\epsilon_{i,1}$, we obtain:

$$\epsilon_i = \epsilon_{i,0}(1 - \rho) + \epsilon_{i,1}\rho. \tag{20}$$

Combining the numerator and denominator, the proportion of minority class samples in $\mathcal{D}_i^h$ is:

$$P(y = 1 \mid (\boldsymbol{x}, y) \in \mathcal{D}_i^h) = \frac{\epsilon_{i,1}\rho}{\epsilon_{i,0}(1 - \rho) + \epsilon_{i,1}\rho}. \tag{21}$$

To establish that the minority class is overrepresented in the misclassified set compared to the original dataset (i.e., $P(y = 1 \mid (\boldsymbol{x}, y) \in \mathcal{D}_i^h) > \rho$), we proceed with the following inequality:

$$\frac{\epsilon_{i,1}\rho}{\epsilon_{i,0}(1 - \rho) + \epsilon_{i,1}\rho} > \rho. \tag{22}$$

Inequality Transformation. Multiply both sides by the positive denominator $\epsilon_{i,0}(1 - \rho) + \epsilon_{i,1}\rho$:

$$\epsilon_{i,1}\rho > \rho \left[\epsilon_{i,0}(1 - \rho) + \epsilon_{i,1}\rho\right]. \tag{23}$$

*Division by $\rho$*: Since $\rho > 0$ (as it is a probability), divide both sides by $\rho$:

$$\epsilon_{i,1} > \epsilon_{i,0}(1 - \rho) + \epsilon_{i,1}\rho. \tag{24}$$

*Rearrangement*: Move terms involving $\epsilon_{i,1}$ to one side:

$$\epsilon_{i,1} - \epsilon_{i,1}\rho > \epsilon_{i,0}(1 - \rho). \tag{25}$$

*Factorization*: Factor out $\epsilon_{i,1}$ on the left-hand side:

$$\epsilon_{i,1}(1 - \rho) > \epsilon_{i,0}(1 - \rho). \tag{26}$$

*Division by $1 - \rho$*: Since $1 - \rho > 0$ (as $\rho < 0.5$), divide both sides by $1 - \rho$:

$$\epsilon_{i,1} > \epsilon_{i,0}. \tag{27}$$

This inequality holds because, in imbalanced datasets where $\rho \ll 0.5$, weak classifiers $\boldsymbol{\pi}_i^{(t+1)}$ trained on $\mathcal{D}_i^{(t)}$ are biased toward the majority class due to the scarcity of minority class samples. This bias results in a higher misclassification rate for the minority class $\epsilon_{i,1}$ compared to the majority class $\epsilon_{i,0}$. Thus, $P(y = 1 \mid (\boldsymbol{x}, y) \in \mathcal{D}_i^h) > \rho$, confirming that the misclassified sample set $\mathcal{D}_i^h$ contains a disproportionately high number of minority class instances compared to the original dataset.

## B.1 ENHANCED MECHANISM EFFICACY

The dynamic enhancement mechanism augments the training subset $\mathcal{D}_i^{(t)}$ with the misclassified sample set $\mathcal{D}_i^h$, forming the updated training subset $\mathcal{D}_i^{(t)} = \mathcal{D}_i^{(t-1)} \cup \mathcal{D}_i^h$. To rigorously demonstrate the efficacy of this mechanism, we analyze the change in the minority class proportion within the training subset.

*Initial Setup*: Let the size of the original training subset be $|\mathcal{D}_i^{(t-1)}| = n$, with the number of minority class samples given by $n \cdot \rho$, where $\rho = P(y = 1)$ is the minority class proportion in $\mathcal{D}_i^{(t-1)}$. The size of the misclassified sample set is $|\mathcal{D}_i^h| = n_h$, and the number of minority class samples in $\mathcal{D}_i^h$ is $n_h \cdot P(y = 1 \mid (\boldsymbol{x}, y) \in \mathcal{D}_i^h)$.

*New Minority Class Proportion*: The updated minority class proportion $\rho'$ in $\mathcal{D}_i^{(t)}$ is:

$$\rho' = \frac{n \cdot \rho + n_h \cdot P(y = 1 \mid (\boldsymbol{x}, y) \in \mathcal{D}_i^h)}{n + n_h}. \tag{28}$$

*Inequality Derivation*: From the previous subsection, we established that $P(y = 1 \mid (\boldsymbol{x}, y) \in \mathcal{D}_i^h) > \rho$. Let $P(y = 1 \mid (\boldsymbol{x}, y) \in \mathcal{D}_i^h) = \rho + \delta$, where $\delta > 0$ represents the excess proportion of minority class samples in $\mathcal{D}_i^h$. Substitute this into the expression for $\rho'$:

$$\rho' = \frac{n \cdot \rho + n_h \cdot (\rho + \delta)}{n + n_h}. \tag{29}$$

*Numerator Expansion*: Expand the numerator:

$$n \cdot \rho + n_h \cdot (\rho + \delta) = n \cdot \rho + n_h \cdot \rho + n_h \cdot \delta = (n + n_h) \cdot \rho + n_h \cdot \delta. \tag{30}$$

*Division by Denominator*: Divide each term by the denominator $n + n_h$:

$$\rho' = \frac{(n + n_h) \cdot \rho}{n + n_h} + \frac{n_h \cdot \delta}{n + n_h} = \rho + \frac{n_h \cdot \delta}{n + n_h}. \tag{31}$$

*Conclusion of Inequality*: Since $n_h > 0$ (as $\mathcal{D}_i^h$ contains misclassified samples) and $\delta > 0$ (due to the higher minority class proportion in $\mathcal{D}_i^h$), the term $\frac{n_h \cdot \delta}{n + n_h} > 0$. Therefore:

$$\rho' > \rho. \tag{32}$$

This increase in the minority class proportion $\rho'$ indicates that the updated training subset $\mathcal{D}_i^{(t)}$ is more balanced than the original $\mathcal{D}_i^{(t-1)}$.

*Error Rate Improvement*: The retrained classifier $h_i'$ on $\mathcal{D}_i^{(t)}$ benefits from the increased representation of minority class samples. The training error rate for the minority class, $\epsilon_{i,1}' = P(\boldsymbol{\pi}_i^{(t+1)}(\boldsymbol{x}) \neq 1 \mid y = 1)$, is influenced by the loss function, which now assigns greater weight to minority class misclassifications due to their higher proportion. According to statistical learning theory, an increase in the number of training samples for a given class reduces the variance of the classifier's predictions for that class. Let the empirical risk for $h_i'$ be:

$$\hat{R}(h_i') = \frac{1}{|\mathcal{D}_i^{(t)}|} \sum_{(\boldsymbol{x}, y) \in \mathcal{D}_i^{(t)}} \mathbb{I}(\boldsymbol{\pi}_i^{t+1}(\boldsymbol{x}) \neq y). \tag{33}$$

The generalization error bound, based on the increased sample size and balanced distribution, is:

$$\epsilon'_{i,1} \leq \hat{\epsilon}'_{i,1} + \mathcal{O}\left(\sqrt{\frac{\ln(1/\delta)}{|\mathcal{D}_i^{(t)}|}}\right), \tag{34}$$

where $\hat{\epsilon}'_{i,1}$ is the empirical error rate for the minority class. As $|\mathcal{D}_i^{(t)}|$ increases and $\rho'$ approaches a more balanced value, $\hat{\epsilon}'_{i,1}$ decreases, leading to $\epsilon'_{i,1} < \epsilon_{i,1}$.

### B.2    Special Scenario Superiority

In scenarios of extreme class imbalance (e.g., $\rho \ll 0.01$), the decision boundary of a classifier trained on $\mathcal{D}_i^{(t-1)}$ tends to shift toward the minority class to minimize the overall error rate, which is dominated by the majority class. This shift causes even "easy" minority class samples—those with distinct features and far from the decision boundary in a balanced setting—to be misclassified as majority class samples. These misclassified samples are identified as "hard-to-classify" due to the imbalance-induced bias.

The dynamic enhancement mechanism addresses this issue by incorporating $\mathcal{D}_i^h$ into $\mathcal{D}_i^{(t)}$. Since $\mathcal{D}_i^h$ is enriched with minority class samples (as $P(y = 1 \mid (\boldsymbol{x}, y) \in \mathcal{D}_i^h) > \rho$), the retrained classifier $h'_i$ on $\mathcal{D}_i^{(t)}$ experiences an improved minority class representation. This adjustment allows $h'_i$ to better capture the minority class's feature distribution. Mathematically, the decision boundary shift can be modeled in a linear classifier context, where the decision function is $f(\boldsymbol{x}) = \boldsymbol{w}^T\boldsymbol{x} + b$. In extreme imbalance, $\boldsymbol{w}$ and $b$ are adjusted to minimize:

$$\frac{1}{M}\sum_{i=1}^{M}\mathbb{I}(f(\boldsymbol{x}_i)y_i < 0), \tag{35}$$

favoring the majority class. The inclusion of $\mathcal{D}_i^h$ increases the penalty for misclassifying minority samples, shifting the boundary back. Consequently, the minority class error rate $\epsilon'_{i,1}$ decreases, and the misclassification of "easy" minority samples is mitigated, demonstrating the superiority of EDEL in extreme imbalance scenarios.

### B.3    Integrated Learning and Overall Performance Enhancement

Building on the convergence analysis of individual weak classifiers in Section 3.2, this section analyzes the integrated classifier's performance and overall error reduction.

This section demonstrates how the EDEL algorithm integrates multiple weak classifiers to enhance overall performance, leveraging classical ensemble learning theories. The integration process combines the outputs of dynamically enhanced weak classifiers to form a robust final classifier, while the iterative refinement of average empirical error ensures convergence to an optimal performance level.

#### B.3.1    Integrated Mechanism for Performance Boost

The final classifier in the EDEL algorithm is defined as $H(\boldsymbol{x}) = \text{sign}\left(\sum_{i=1}^{N}\alpha_i h'_i(\boldsymbol{x})\right)$, where each $h'_i$ is a weak classifier trained on the dynamically enhanced subset $\mathcal{D}_i^{(t)} = \mathcal{D}_i^{(t-1)} \cup \mathcal{D}_i^h$ (as specified in the pipeline), and $\alpha_i$ represents the weight assigned to the $i$-th weak classifier. The weight $\alpha_i$ is determined based on the classifier's performance, specifically using the formula $\alpha_i = \frac{1}{2}\ln\frac{1-\epsilon_i}{\epsilon_i}$, where $\epsilon_i = P(h'_i(\boldsymbol{x}) \neq y)$ is the error rate of $h'_i$ over the true distribution $\mathcal{D}$. The error rate $\epsilon_i$ is ensured to be less than 0.5 due to the dynamic enhancement mechanism, which increases the minority class proportion in $\mathcal{D}_i^{(t)}$, thereby improving the classifier's ability to handle imbalanced data.

To quantify the performance boost, we derive the upper bound on the final classifier's error rate $\epsilon_{\text{final}} = P(H(\boldsymbol{x}) \neq y)$ using classical AdaBoost theory (Freund & Schapire, 1997a). The error rate of the integrated classifier can be bounded as follows:

$$\epsilon_{\text{final}} = P\left(\text{sign}\left(\sum_{i=1}^{N}\alpha_i h_i'(\boldsymbol{x})\right) \neq y\right). \tag{36}$$

Consider the weighted margin $S(\boldsymbol{x}) = \sum_{i=1}^{N}\alpha_i h_i'(\boldsymbol{x})$. The classification error occurs when $y \cdot S(\boldsymbol{x}) \leq 0$. The expected error can be analyzed using the exponential loss framework, where the upper bound is given by:

$$\epsilon_{\text{final}} \leq \prod_{i=1}^{N}\sqrt{4\epsilon_i(1-\epsilon_i)}. \tag{37}$$

To derive this bound, note that for each weak classifier $h_i'$, the weighted contribution to the margin is influenced by its error rate $\epsilon_i$. The probability of correct classification by $h_i'$ is $1 - \epsilon_i$, and the AdaBoost weighting scheme ensures that the margin grows with the number of classifiers. The term $\sqrt{4\epsilon_i(1-\epsilon_i)}$ arises from the analysis of the exponential loss $\exp(-y \cdot S(\boldsymbol{x}))$, where the error probability is bounded by the product of individual classifier contributions. Since $\epsilon_i < 0.5$, the maximum value of $4\epsilon_i(1-\epsilon_i)$ occurs at $\epsilon_i = 0.5$, yielding 1, and for $\epsilon_i < 0.5$, $4\epsilon_i(1-\epsilon_i) < 1$. Thus:

$$\sqrt{4\epsilon_i(1-\epsilon_i)} < 1. \tag{38}$$

As $N$ increases, the product $\prod_{i=1}^{N}\sqrt{4\epsilon_i(1-\epsilon_i)}$ decreases exponentially because each factor is less than 1. This exponential decay demonstrates that the error rate $\epsilon_{\text{final}}$ diminishes with the number of weak classifiers, proving a significant performance boost through integration.

The diversity among weak classifiers $h_i'$ further enhances this improvement. This diversity stems from the stratified sampling of $\mathcal{D}_i^{(t-1)}$ and the dynamic enhancement with $\mathcal{D}_i^h$, which introduces variations in the training data across different classifiers. The reduced correlation among $h_i'$ ensures that their errors are not perfectly aligned, leading to a more robust ensemble. This diversity effect lowers the overall $\epsilon_{\text{final}}$ beyond the theoretical bound, reinforcing the efficacy of the integrated mechanism.

### B.3.2 CONVERGENCE OF AVERAGE EMPIRICAL ERROR VIA MCDIARMID INEQUALITY

To ensure the iterative optimization process of EDEL converges to an optimal performance level, we analyze the convergence of the average empirical error using the McDiarmid inequality (McDiarmid, 1989). The average empirical error at iteration $t$ is defined as:

$$\hat{E}_t = \frac{1}{n}\sum_{i=1}^{n}\hat{\epsilon}_i^{(t)}, \tag{39}$$

where $n$ is the number of weak classifiers, and the empirical error for the $i$-th weak classifier is:

$$\hat{\epsilon}_i^{(t+1)} = \frac{1}{|\mathcal{D}_i^{(t)}|}\sum_{(\boldsymbol{x},y)\in\mathcal{D}_i^{(t)}}\mathbb{I}(\boldsymbol{\pi}_i^{(t+1)}(\boldsymbol{x}) \neq y), \tag{40}$$

with $|\mathcal{D}_i^{(t)}|$ denoting the size of the training set $\mathcal{D}_i^{(t)}$ at iteration $t$. Our goal is to prove that the average empirical error in the next iteration $\hat{E}_{t+1}$ is likely to decrease by at least $\Delta E > 0$, with a probability bound given by:

$$P(\hat{E}_{t+1} \leq \hat{E}_t - \Delta E) \geq 1 - 2\exp(-2\Delta E^2 n). \tag{41}$$

*Bounded Difference Condition*: We treat $\hat{E}_{t+1}$ as a function of the training sets for all weak classifiers, defined as $f(\mathcal{D}_1^{(t+1)}, \ldots, \mathcal{D}_n^{(t+1)}) = \hat{E}_{t+1}$. To apply the McDiarmid inequality, we need to

verify the bounded difference condition. Consider changing the training set $\mathcal{D}_i^{(t+1)}$ to $\mathcal{D}_i'^{(t+1)}$ for a fixed $i$. This change affects only the empirical error $\hat{\epsilon}_i^{(t+1)}$ of the $i$-th weak classifier, while $\hat{\epsilon}_j^{(t+1)}$ (for $j \neq i$) remains unchanged. The difference in the function value is:

$$|f(\mathcal{D}_1^{(t+1)}, \ldots, \mathcal{D}_i^{(t+1)}, \ldots, \mathcal{D}_n^{(t+1)}) - f(\mathcal{D}_1^{(t+1)}, \ldots, \mathcal{D}_i'^{(t+1)}, \ldots, \mathcal{D}_n^{(t+1)})| = \frac{1}{n}|\hat{\epsilon}_i^{(t+1)} - \hat{\epsilon}_i'^{(t+1)}|. \tag{42}$$

Since $\hat{\epsilon}_i^{(t+1)}$ is a misclassification rate ranging from 0 to 1, the maximum possible change when altering the entire training set $\mathcal{D}_i^{(t+1)}$ is from 0 to 1 (or vice versa). Thus:

$$|\hat{\epsilon}_i^{(t+1)} - \hat{\epsilon}_i'^{(t+1)}| \leq 1, \tag{43}$$

This establishes the bounded difference constant $c_i = \frac{1}{n}$ for each $i$. The sum of squared bounded differences is:

$$\sum_{i=1}^{n} c_i^2 = \sum_{i=1}^{n} \left(\frac{1}{n}\right)^2 = n \cdot \frac{1}{n^2} = \frac{1}{n}. \tag{44}$$

*McDiarmid Inequality Application*: The McDiarmid inequality (McDiarmid, 1989) states that for a function $Y = f(X_1, \ldots, X_n)$ satisfying the bounded difference condition, the probability of deviating from its expectation is bounded by:

$$P(|Y - \mathbb{E}[Y]| \geq t) \leq 2 \exp\left(-\frac{2t^2}{\sum_{i=1}^{n} c_i^2}\right). \tag{45}$$

Substituting the sum of squared bounded differences:

$$P(|\hat{E}_{t+1} - \mathbb{E}[\hat{E}_{t+1}]| \geq t) \leq 2 \exp\left(-\frac{2t^2}{\frac{1}{n}}\right) = 2 \exp(-2t^2 n). \tag{46}$$

*Error Reduction Analysis*: To prove $\hat{E}_{t+1} \leq \hat{E}_t - \Delta E$, we assume that the expected average empirical error after the update satisfies $\mathbb{E}[\hat{E}_{t+1}] \leq \hat{E}_t - \Delta E$, where $\Delta E > 0$ represents the expected reduction in error due to the inclusion of misclassified samples $\mathcal{D}_i^h$. This assumption is based on the intuition that focusing on hard-to-classify samples improves the classifier's performance. We need to bound the probability that $\hat{E}_{t+1}$ exceeds $\hat{E}_t - \Delta E$.

Set $t = \Delta E$ in the inequality:

$$P(\hat{E}_{t+1} - \mathbb{E}[\hat{E}_{t+1}] \geq \Delta E) \leq \exp(-2\Delta E^2 n). \tag{47}$$

This represents the probability that $\hat{E}_{t+1}$ is $\Delta E$ above its expectation. Since $\mathbb{E}[\hat{E}_{t+1}] \leq \hat{E}_t - \Delta E$, the event $\hat{E}_{t+1} > \hat{E}_t - \Delta E$ occurs if $\hat{E}_{t+1} - \mathbb{E}[\hat{E}_{t+1}] > \hat{E}_t - \Delta E - \mathbb{E}[\hat{E}_{t+1}]$. Given $\hat{E}_t - \Delta E - \mathbb{E}[\hat{E}_{t+1}] \geq 0$, the maximum deviation is bounded by $\Delta E$. Thus:

$$P(\hat{E}_{t+1} > \hat{E}_t - \Delta E) \leq \exp(-2\Delta E^2 n). \tag{48}$$

For a conservative estimate, considering both tails of the distribution (as $\hat{E}_{t+1}$ could deviate in either direction), the double-sided bound is:

$$P(|\hat{E}_{t+1} - \mathbb{E}[\hat{E}_{t+1}]| \geq \Delta E) \leq 2 \exp(-2\Delta E^2 n). \tag{49}$$

Therefore, the probability that $\hat{E}_{t+1}$ does not exceed $\hat{E}_t - \Delta E$ is:

$$P(\hat{E}_{t+1} \leq \hat{E}_t - \Delta E) \geq 1 - 2\exp(-2\Delta E^2 n). \tag{50}$$

*Convergence Property*: As the number of weak classifiers $n$ increases, the exponent $-2\Delta E^2 n$ becomes more negative, causing the probability bound $2\exp(-2\Delta E^2 n)$ to approach 0. This implies that $\hat{E}_{t+1} \leq \hat{E}_t - \Delta E$ holds with probability approaching 1. Over multiple iterations, the average empirical error $\hat{E}_t$ converges to a limiting value $E_\infty$, which represents the optimal average error achievable given the data distribution and the capacity of the weak classifiers.

This convergence is driven by the iterative process of EDEL, where the inclusion of $\mathcal{D}_i^h$ refines the training sets $\mathcal{D}_i^{(t+1)}$, allowing the weak classifiers to better capture the underlying patterns, particularly for minority class instances. The McDiarmid inequality thus provides a statistical guarantee that the algorithm's performance improves iteratively, enhancing the overall efficacy of the ensemble.

## C EXPERIMENTAL DETAILS

### C.1 DATASET AND METRIC

**Dataset.** These datasets were chosen because they represent real-world scenarios where class imbalance is prevalent and poses significant challenges for traditional machine learning models. By evaluating our method on these diverse datasets, we ensure its generalizability and effectiveness in handling various imbalance scenarios. Below are the details of each dataset:

- **Spambase Dataset (SBD)** [1]: This dataset contains 4,601 samples used to classify emails as spam (positive class) or non-spam. With an IR of 1.54, it represents a moderately imbalanced scenario. Spam detection is critical in email filtering systems to reduce user inconvenience caused by unwanted messages.

- **Adult Income Dataset (AID)** [2]: This dataset includes 48,842 samples to predict whether an individual's income exceeds 50K per year (positive class). The IR is 3.18, indicating a moderate imbalance. This dataset is often used to study socioeconomic factors influencing income levels, where accurately identifying high-income individuals is key.

- **Taiwan Credit Card Default (TCD)** [3]: This dataset has 30,000 samples to predict whether a credit card holder will default on payments (positive class). With an IR of 3.52, it reflects a typical credit risk assessment scenario where identifying potential defaulters is crucial for financial institutions.

- **CDC Diabetes Health Indicators (CDH)** [4]: This dataset consists of 253,680 samples used to predict whether an individual has diabetes (positive class) based on health indicators. The IR is 6.18, highlighting the challenge of diagnosing rare diseases in large populations.

- **Bank Marketing Dataset (BMD)** [5]: This dataset includes 41,188 samples for predicting whether a client will subscribe to a term deposit (positive class). With an IR of 7.88, it represents a marketing campaign scenario where successful subscriptions are relatively rare.

- **Give Me Some Credit Dataset (GMSC)** [6]: This dataset consists of 150,000 samples to predict whether an individual will experience financial distress in the next two years (positive class). The IR is 13.96, emphasizing the difficulty of predicting rare but high-impact events in credit risk modeling.

- **Credit Card Fraud Detection (CCFD)** [7]: This dataset includes 284,807 samples for detecting whether a transaction is fraudulent (positive class). With an extreme IR of 577.88, it poses signif-

---

[1] https://archive.ics.uci.edu/ml/datasets/spambase
[2] https://archive.ics.uci.edu/ml/datasets/adult
[3] https://archive.ics.uci.edu/ml/datasets/default+of+credit+card+clients
[4] https://www.kaggle.com/datasets/alexteboul/diabetes-health-indicators-dataset/data
[5] https://archive.ics.uci.edu/ml/datasets/Bank+Marketing
[6] https://www.kaggle.com/c/GiveMeSomeCredit
[7] https://www.kaggle.com/mlg-ulb/creditcardfraud

icant challenges for fraud detection systems, where missing a fraudulent transaction can lead to substantial losses.

**Evaluation Metrics.** We employ AUC and F-measure as evaluation metrics to assess the performance of the EDEL algorithm in imbalanced classification tasks. These metrics are particularly suitable for high-stakes applications such as fraud detection and credit risk assessment, where class imbalance poses significant challenges.

**AUC** is the area under the Receiver Operating Characteristic (ROC) curve, which effectively reflects the overall accuracy of the classifier in distinguishing between positive and negative classes across all possible thresholds. It ranges from 0 to 1, with 1 indicating perfect classification and 0.5 representing random guessing. The AUC is calculated based on the True Positive Rate (TPR) and False Positive Rate (FPR), defined as:

$$\text{TPR} = \frac{\text{TP}}{\text{TP} + \text{FN}}, \quad \text{FPR} = \frac{\text{FP}}{\text{FP} + \text{TN}},$$

where TP (True Positives) is the number of correctly predicted positive instances, FN (False Negatives) is the number of positive instances incorrectly predicted as negative, FP (False Positives) is the number of negative instances incorrectly predicted as positive, and TN (True Negatives) is the number of correctly predicted negative instances.

**F-measure** is obtained using the formula:

$$F\text{-measure} = \frac{(1 + \beta^2) \cdot \text{Recall} \cdot \text{Precision}}{\beta^2 \cdot \text{Precision} + \text{Recall}},$$

where $\text{Recall} = \frac{\text{TP}}{\text{TP+FN}}$, $\text{Precision} = \frac{\text{TP}}{\text{TP+FP}}$, and $\beta$ is a factor used to balance and weight the importance of Recall and Precision. When $\beta > 1$, Recall has a greater impact; when $\beta < 1$, Precision has a greater impact. In this study, we use the F1-measure, where $\beta$ is set to 1, giving equal weight to Precision and Recall. The F1-measure ranges from 0 to 1, with higher values indicating better balance between identifying positive instances and avoiding false positives.

## C.2 EXPERIMENTAL SETTING

**Baselines.** We evaluate EDEL against five baselines, including:

(1) **basic method**, the Original instances with no imbalance handling applied;

(2) **data-level methods:**

- SMOTE (Chawla et al., 2002), which generates synthetic minority samples via interpolation between existing minority instances.
- RandomUnderSampler (RUS) (He & Garcia, 2009), which randomly selects majority class instances to match the minority class size.
- SMOTE-TLNN-DEPSO (Dixit & Mani, 2023), which integrates Two-Layer Nearest Neighbor and Differential Evolution Particle Swarm Optimization to handle noisy data.

(3) **ensemble learning methods:**

- MESA (Liu et al., 2020), which employs a meta-sampler trained via soft actor-critic reinforcement learning to learn an adaptive under-sampling strategy; it iteratively computes error distributions on training and validation sets to form a meta-state, uses the meta-sampler to output a parameter $\mu$ for a Gaussian function that assigns sampling weights based on classification errors, and builds a cascade ensemble of classifiers on balanced subsets, optimizing for generalization performance; configured in our experiments with $random\_state = 42$, while keeping hyperparameters (e.g., $metric = aucprc$, $max\_estimators = 10$, $num\_bins = 5$, $\sigma = 0.2$, $train\_ir = 1$, $update\_steps = 1000$, $start\_steps = 500$, $hidden\_size = 50$) at their default values.

- CHRE (Zhao et al., 2025), a hybrid resampling method, synthesizes new minority samples based on sample contribution using Euclidean distance to balance information and noise levels, configured in our experiments with hyperparameters $\lambda_1 = 0.8$, $\lambda_2 = 0.5$, $K = 5$, and a maximum of 10 iterations.

All methods, including EDEL and the baselines, are evaluated using four classifiers: Decision Tree, Random Forest, XGBoost, and LightGBM. Experimental results are reported by 5-fold stratified cross-validation, with each fold allocating 80% for training and 20% for testing.

**Classifier parameters Detailed**. Classifier parameters are set as follows: Decision-TreeClassifier with criterion=$'gini'$, max_depth=$None$, min_samples_split=2, min_samples_leaf=1, and max_features=$None$; RandomForestClassifier with n_estimators=100, criterion=$'gini'$, max_depth=$None$, min_samples_split=2, min_samples_leaf=1, and max_features=$'sqrt'$; XGBClassifier with n_estimators=100, max_depth=6, learning_rate=0.3, and objective=$'binary : logistic'$; and LightGBM with boosting_type=$'gbdt'$, num_leaves=31, max_depth=$-1$, learning_rate=0.1, and n_estimators=100. These settings ensure reproducibility of experimental results.

# D DISCUSSION

## D.1 SENSITIVITY ANALYSIS

To evaluate the impact of the number of weak classifiers $N$ on the performance of Error-Driven Ensemble Learning (EDEL), we conduct a sensitivity analysis across seven real-world datasets: Spambase (SBD, IR=1.54), Adult Income (AID, IR=3.18), Taiwan Credit Card Default (TCD, IR=3.52), CDC Diabetes Health Indicators (CDH, IR=6.18), Bank Marketing (BMD, IR=7.88), Give Me Some Credit (GMSC, IR=13.96), and Credit Card Fraud Detection (CCFD, IR=577.88). We vary $N$ from 1 to 5, where EDEL (1) represents the baseline without EDEL's dynamic augmentation. Four classifiers are evaluated: Decision Tree (DT), Random Forest (RF), XGBoost (XGB), and LightGBM (LGBM), using AUC and F1-measure as metrics (Section C.1).

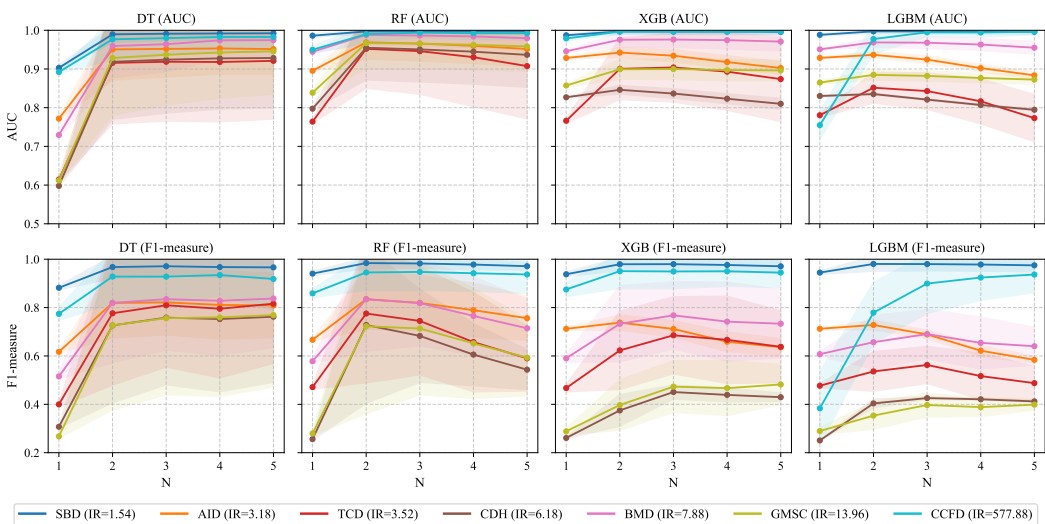

Figure 3: Sensitivity analysis of EDEL performance with varying number of weak classifiers ($N$) across seven datasets. Top row: AUC for DT, RF, XGB, and LGBM. Bottom row: F1-measure.

Figure 3 illustrates EDEL's performance trends across datasets and classifiers(detailed data in Table 7 and Table 8). On CCFD (IR=577.88), EDEL with LGBM achieves an AUC of 0.9948 and F1-measure of 0.8993 at $N = 3$, compared to 0.7549 and 0.3834 at $N = 1$, reflecting a 31.7% AUC increase and 135% F1 improvement. Similarly, on GMSC (IR=13.96) with DT, F1 rises from 0.2679 ($N = 1$) to 0.7690 ($N = 5$), a 187% gain. For moderate imbalance, such as SBD (IR=1.54) with RF, AUC improves from 0.9860 ($N = 1$) to 0.9972 ($N = 3$). Performance typically peaks at

$N = 3$, with diminishing returns beyond this point, likely due to overlapping misclassified instances reducing subset diversity (Section B.3).

Table 7: AUC sensitivity analysis results for EDEL across datasets with varying number of weak classifiers ($N$). EDEL (1) represents the baseline ($N = 1$, no EDEL).

| Clf | $N$ | SBD | AID | TCD | CDH | BMD | GMSC | CCFD |
|---|---|---|---|---|---|---|---|---|
| DT | EDEL (1) | 0.9030 ± 0.0075 | 0.7717 ± 0.0041 | 0.6143 ± 0.0074 | 0.5979 ± 0.0031 | 0.7294 ± 0.0066 | 0.6120 ± 0.0047 | 0.8921 ± 0.0041 |
| | EDEL (2) | 0.9896 ± 0.0211 | 0.9506 ± 0.0796 | 0.9158 ± 0.1578 | 0.9186 ± 0.1488 | 0.9594 ± 0.0822 | 0.9283 ± 0.1476 | 0.9767 ± 0.0521 |
| | EDEL (3) | 0.9911 ± 0.0182 | 0.9519 ± 0.0724 | 0.9187 ± 0.1536 | 0.9241 ± 0.1381 | 0.9641 ± 0.0705 | 0.9369 ± 0.1300 | 0.9797 ± 0.0454 |
| | EDEL (4) | 0.9918 ± 0.0162 | 0.9531 ± 0.0661 | 0.9181 ± 0.1546 | 0.9275 ± 0.1312 | 0.9740 ± 0.0500 | 0.9425 ± 0.1184 | 0.9827 ± 0.0386 |
| | EDEL (5) | 0.9923 ± 0.0148 | 0.9514 ± 0.0644 | 0.9210 ± 0.1509 | 0.9284 ± 0.1292 | 0.9745 ± 0.0475 | 0.9457 ± 0.1107 | 0.9827 ± 0.0387 |
| RF | EDEL (1) | 0.9860 ± 0.0041 | 0.8954 ± 0.0036 | 0.7639 ± 0.0046 | 0.7974 ± 0.0025 | 0.9444 ± 0.0024 | 0.8387 ± 0.0055 | 0.9497 ± 0.0108 |
| | EDEL (2) | 0.9971 ± 0.0062 | 0.9688 ± 0.0431 | 0.9521 ± 0.1020 | 0.9549 ± 0.0833 | 0.9881 ± 0.0246 | 0.9669 ± 0.0679 | 0.9900 ± 0.0224 |
| | EDEL (3) | 0.9972 ± 0.0058 | 0.9656 ± 0.0448 | 0.9461 ± 0.1117 | 0.9506 ± 0.0841 | 0.9865 ± 0.0271 | 0.9656 ± 0.0657 | 0.9939 ± 0.0137 |
| | EDEL (4) | 0.9971 ± 0.0060 | 0.9598 ± 0.0454 | 0.9305 ± 0.1280 | 0.9447 ± 0.0840 | 0.9842 ± 0.0287 | 0.9625 ± 0.0653 | 0.9925 ± 0.0167 |
| | EDEL (5) | 0.9970 ± 0.0057 | 0.9516 ± 0.0463 | 0.9075 ± 0.1364 | 0.9365 ± 0.0839 | 0.9795 ± 0.0306 | 0.9583 ± 0.0641 | 0.9923 ± 0.0172 |
| XGB | EDEL (1) | 0.9872 ± 0.0031 | 0.9285 ± 0.0025 | 0.7661 ± 0.0059 | 0.8269 ± 0.0025 | 0.9458 ± 0.0018 | 0.8573 ± 0.0043 | 0.9789 ± 0.0062 |
| | EDEL (2) | 0.9971 ± 0.0060 | 0.9424 ± 0.0127 | 0.9004 ± 0.0795 | 0.8462 ± 0.0113 | 0.9756 ± 0.0193 | 0.8992 ± 0.0258 | 0.9966 ± 0.0076 |
| | EDEL (3) | 0.9968 ± 0.0065 | 0.9343 ± 0.0160 | 0.9039 ± 0.0897 | 0.8365 ± 0.0125 | 0.9761 ± 0.0206 | 0.8999 ± 0.0270 | 0.9965 ± 0.0077 |
| | EDEL (4) | 0.9968 ± 0.0063 | 0.9179 ± 0.0164 | 0.8931 ± 0.1001 | 0.8231 ± 0.0131 | 0.9745 ± 0.0227 | 0.8973 ± 0.0276 | 0.9964 ± 0.0079 |
| | EDEL (5) | 0.9960 ± 0.0072 | 0.9025 ± 0.0166 | 0.8739 ± 0.1096 | 0.8100 ± 0.0141 | 0.9708 ± 0.0244 | 0.8963 ± 0.0270 | 0.9968 ± 0.0071 |
| LGBM | EDEL (1) | 0.9882 ± 0.0035 | 0.9287 ± 0.0029 | 0.7805 ± 0.0052 | 0.8302 ± 0.0030 | 0.9508 ± 0.0019 | 0.8649 ± 0.0039 | 0.7549 ± 0.0368 |
| | EDEL (2) | 0.9973 ± 0.0056 | 0.9366 ± 0.0079 | 0.8516 ± 0.0399 | 0.8352 ± 0.0024 | 0.9687 ± 0.0123 | 0.8849 ± 0.0138 | 0.9774 ± 0.0388 |
| | EDEL (3) | 0.9969 ± 0.0065 | 0.9243 ± 0.0100 | 0.8429 ± 0.0474 | 0.8209 ± 0.0021 | 0.9679 ± 0.0152 | 0.8821 ± 0.0130 | 0.9948 ± 0.0116 |
| | EDEL (4) | 0.9969 ± 0.0062 | 0.9023 ± 0.0098 | 0.8165 ± 0.0584 | 0.8070 ± 0.0021 | 0.9632 ± 0.0155 | 0.8768 ± 0.0125 | 0.9944 ± 0.0126 |
| | EDEL (5) | 0.9965 ± 0.0069 | 0.8836 ± 0.0090 | 0.7731 ± 0.0609 | 0.7946 ± 0.0027 | 0.9550 ± 0.0166 | 0.8726 ± 0.0121 | 0.9965 ± 0.0079 |

Table 8: F1-measure sensitivity analysis results for EDEL across datasets with varying number of weak classifiers ($N$). EDEL (1) represents the baseline ($N = 1$, no EDEL).

| Clf | $N$ | SBD | AID | TCD | CDH | BMD | GMSC | CCFD |
|---|---|---|---|---|---|---|---|---|
| DT | EDEL (1) | 0.8819 ± 0.0088 | 0.6171 ± 0.0043 | 0.3999 ± 0.0104 | 0.3072 ± 0.0047 | 0.5155 ± 0.0119 | 0.2679 ± 0.0082 | 0.7741 ± 0.0272 |
| | EDEL (2) | 0.9676 ± 0.0465 | 0.8194 ± 0.1449 | 0.7765 ± 0.2973 | 0.7262 ± 0.3202 | 0.8194 ± 0.2467 | 0.7267 ± 0.3502 | 0.9283 ± 0.1116 |
| | EDEL (3) | 0.9709 ± 0.0412 | 0.8215 ± 0.1339 | 0.8096 ± 0.2554 | 0.7584 ± 0.2778 | 0.8349 ± 0.2103 | 0.7562 ± 0.3172 | 0.9279 ± 0.1106 |
| | EDEL (4) | 0.9675 ± 0.0403 | 0.8110 ± 0.1411 | 0.7952 ± 0.2864 | 0.7526 ± 0.3005 | 0.8282 ± 0.2249 | 0.7594 ± 0.3263 | 0.9345 ± 0.0882 |
| | EDEL (5) | 0.9663 ± 0.0425 | 0.8087 ± 0.1333 | 0.8161 ± 0.2472 | 0.7622 ± 0.2714 | 0.8375 ± 0.2023 | 0.7690 ± 0.2987 | 0.9182 ± 0.1289 |
| RF | EDEL (1) | 0.9406 ± 0.0095 | 0.6669 ± 0.0045 | 0.4711 ± 0.0086 | 0.2561 ± 0.0046 | 0.5781 ± 0.0086 | 0.2796 ± 0.0085 | 0.8588 ± 0.0186 |
| | EDEL (2) | 0.9840 ± 0.0232 | 0.8349 ± 0.1250 | 0.7754 ± 0.2866 | 0.7280 ± 0.3236 | 0.8344 ± 0.2153 | 0.7229 ± 0.3618 | 0.9453 ± 0.0744 |
| | EDEL (3) | 0.9824 ± 0.0209 | 0.8192 ± 0.1143 | 0.7448 ± 0.2229 | 0.6829 ± 0.1925 | 0.8185 ± 0.1752 | 0.7135 ± 0.2748 | 0.9475 ± 0.0768 |
| | EDEL (4) | 0.9779 ± 0.0222 | 0.7893 ± 0.1118 | 0.6574 ± 0.2060 | 0.6057 ± 0.1285 | 0.7656 ± 0.1626 | 0.6520 ± 0.2271 | 0.9420 ± 0.0746 |
| | EDEL (5) | 0.9711 ± 0.0154 | 0.7562 ± 0.0917 | 0.5901 ± 0.1350 | 0.5433 ± 0.0827 | 0.7145 ± 0.1194 | 0.5929 ± 0.1588 | 0.9370 ± 0.0780 |
| XGB | EDEL (1) | 0.9379 ± 0.0094 | 0.7122 ± 0.0066 | 0.4673 ± 0.0157 | 0.2610 ± 0.0030 | 0.5904 ± 0.0159 | 0.2884 ± 0.0128 | 0.8749 ± 0.0298 |
| | EDEL (2) | 0.9792 ± 0.0280 | 0.7378 ± 0.0316 | 0.6230 ± 0.1620 | 0.3749 ± 0.0658 | 0.7330 ± 0.1578 | 0.3972 ± 0.1052 | 0.9506 ± 0.0722 |
| | EDEL (3) | 0.9796 ± 0.0282 | 0.7116 ± 0.0345 | 0.6855 ± 0.1595 | 0.4508 ± 0.0296 | 0.7679 ± 0.1386 | 0.4732 ± 0.1067 | 0.9493 ± 0.0708 |
| | EDEL (4) | 0.9762 ± 0.0301 | 0.6587 ± 0.0407 | 0.6667 ± 0.1806 | 0.4390 ± 0.0290 | 0.7419 ± 0.1638 | 0.4670 ± 0.1136 | 0.9502 ± 0.0621 |
| | EDEL (5) | 0.9707 ± 0.0291 | 0.6361 ± 0.0356 | 0.6380 ± 0.1492 | 0.4296 ± 0.0176 | 0.7333 ± 0.1409 | 0.4821 ± 0.0827 | 0.9446 ± 0.0628 |
| LGBM | EDEL (1) | 0.9447 ± 0.0081 | 0.7124 ± 0.0049 | 0.4770 ± 0.0088 | 0.2506 ± 0.0029 | 0.6074 ± 0.0096 | 0.2895 ± 0.0138 | 0.3834 ± 0.1507 |
| | EDEL (2) | 0.9804 ± 0.0300 | 0.7282 ± 0.0201 | 0.5360 ± 0.0821 | 0.4039 ± 0.0181 | 0.6569 ± 0.1117 | 0.3534 ± 0.0572 | 0.7791 ± 0.1265 |
| | EDEL (3) | 0.9799 ± 0.0280 | 0.6891 ± 0.0180 | 0.5627 ± 0.0769 | 0.4258 ± 0.0058 | 0.6904 ± 0.1018 | 0.3972 ± 0.0495 | 0.8993 ± 0.1237 |
| | EDEL (4) | 0.9780 ± 0.0275 | 0.6217 ± 0.0234 | 0.5169 ± 0.0961 | 0.4209 ± 0.0037 | 0.6541 ± 0.1079 | 0.3884 ± 0.0386 | 0.9241 ± 0.0944 |
| | EDEL (5) | 0.9747 ± 0.0266 | 0.5840 ± 0.0146 | 0.4876 ± 0.0688 | 0.4121 ± 0.0047 | 0.6403 ± 0.0782 | 0.3988 ± 0.0228 | 0.9366 ± 0.0753 |

These results highlight EDEL's effectiveness in enhancing minority class detection, particularly in extreme imbalance scenarios like CCFD, where dynamic incorporation of hard-to-classify instances (Section 2) refines decision boundaries. The performance saturation at $N = 3$ aligns with the theoretical convergence of average empirical error (Section B.3), balancing computational complexity and classification accuracy, making EDEL practical for real-world applications.

## D.2 DIFFERENCES BETWEEN EDEL AND ADABOOST

Although EDEL draws inspiration from AdaBoost (Freund & Schapire, 1997b) in its iterative, error-driven refinement, the two algorithms differ fundamentally in their design, mechanisms, and handling of class imbalance. These distinctions arise from EDEL's focus on explicit data augmentation for imbalanced learning, contrasted with AdaBoost's weight-based boosting for general classification. Below, we outline key differences, incorporating perspectives on ensemble integration and local iterative processes.

**1. Core Mechanism: Sample Handling and Update Strategy.**

- AdaBoost operates on the entire dataset per iteration, multiplicatively updating sample weights to emphasize misclassified instances. For a sample $(\mathbf{x}_i, y_i)$, the update is

$w_i^{(t+1)} = w_i^{(t)} \cdot \exp(-\alpha_t \cdot y_i \cdot h_t(\mathbf{x}_i))$, where $\alpha_t = \frac{1}{2} \ln \frac{1-\epsilon_t}{\epsilon_t}$ and $\epsilon_t$ is the weighted error. This forms a sequential process where each weak learner $h_t$ is trained on a reweighted distribution, implicitly focusing on hard samples via probabilistic resampling.

- EDEL, however, partitions the dataset into $N$ subsets $\{\mathcal{D}_i\}_{i=1}^N$ and explicitly augments each with misclassified samples $\mathcal{D}_i^h$ from the complement $(\mathcal{D} \setminus \mathcal{D}_i)$: $\mathcal{D}_i^{(t)} = \mathcal{D}_i^{(t-1)} \cup \mathcal{D}_i^h$. This additive augmentation creates parallel, diverse views without global reweighting. From an additional perspective, within each subset's iteration, EDEL can be viewed as a 0/1 selection process over the remaining samples—selecting (1) misclassified ones for inclusion and discarding (0) correctly classified ones—forming a localized, binary decision mini-process akin to a simplified expert allocation, but tailored for imbalance correction.

**2. Focus on Class Imbalance and Sample Difficulty.**

- AdaBoost is geared toward general boosting, converting weak learners (accuracy slightly $>0.5$) into strong ones without explicit imbalance handling. It amplifies misclassified samples globally, which may indirectly benefit minorities if they are frequently erred, but lacks guarantees for disproportionate minority focus in extreme imbalance (e.g., IR $>100$).

- EDEL explicitly addresses imbalance by proving (via Bayes' theorem) that $\mathcal{D}_i^h$ enriches minorities ($P(y = 1 \mid (\mathbf{x}, y) \in \mathcal{D}_i^h) > \rho$), leading to iterative balancing ($\rho' > \rho$). It targets "hard-to-classify" samples (Definition 2), which are often minorities, making it superior for high-IR scenarios (e.g., CCFD with IR=577.88), as validated empirically.

**3. Ensemble Integration and Extensibility.**

- AdaBoost integrates sequentially via weighted voting: $H(\mathbf{x}) = \text{sign}(\sum_t \alpha_t h_t(\mathbf{x}))$, with $\alpha_t$ reflecting weak learner strength. This sequential nature limits parallelism.

- EDEL employs uniform probability averaging over parallel weak classifiers: $\Theta = \frac{1}{N} \sum_i \hat{\Theta}_i$, supporting efficient computation. From an extended perspective, this uniform averaging can be generalized to weighted averaging (e.g., $\Theta = \sum_i \alpha_i \hat{\Theta}_i$, where $\alpha_i$ could be derived from subset-specific errors), allowing flexibility for future adaptations while maintaining focus on imbalance.

**4. Theoretical Foundations and Convergence.**

- AdaBoost's bounds emphasize exponential error decay: training error $\leq \exp(-2\sum_t \gamma_t^2)$, with $\gamma_t = \frac{1}{2} - \epsilon_t$, rooted in exponential loss and VC-dimension for generalization.

- EDEL uses stratified sampling (Chebyshev for representativeness), Rademacher complexity for generalization, and McDiarmid's inequality for convergence ($P(|\hat{\epsilon} - \mathbb{E}[\hat{\epsilon}]| \geq \delta) \leq 2\exp(-2\delta^2 n)$). Its proofs target imbalance-specific properties, like minority enrichment, differing from AdaBoost's margin-based analysis.

In essence, AdaBoost is a general booster via sequential weighting, while EDEL is an imbalance-specialized ensemble through parallel augmentation and selective inclusion. These differences enable EDEL's superior performance in imbalanced domains, as shown in experiments, without requiring prior knowledge of weak learner accuracies.

# E  RELATED WORKS

Existing approaches to address class imbalance are categorized into three groups: data-level methods, algorithm-level methods, and ensemble learning methods, each offering distinct strategies to address this issue.

**Data-Level Methods** Data-level methods seek to rebalance class distributions through dataset modification, providing classifier-agnostic solutions that enhance generalizability across models. The Synthetic Minority Over-sampling Technique (SMOTE) (Chawla et al., 2002) generates synthetic minority samples via interpolation, inspiring variants that address its over-generalization tendencies. For instance, Borderline-SMOTE (Han et al., 2005) targets samples near decision boundaries,

while ADASYN (He et al., 2008) adaptively generates samples based on local density. Safe-Level-SMOTE (Bunkhumpornpat et al., 2009) ensures synthetic samples remain within safe minority class regions, and FW-SMOTE (Maldonado et al., 2022) incorporates feature weighting to improve sample relevance. RCSMOTE (Soltanzadeh & Hashemzadeh, 2021) refines SMOTE by constraining synthetic sample ranges using distances to majority class samples, reducing noise and class overlap, particularly effective for moderately imbalanced datasets. Similarly, MLBOITE (Teng et al., 2024) extends SMOTE to multi-label datasets with a three-phase framework (seed set construction, border-line sample resampling, and internal sample resampling), optimizing label assignment for complex multi-label scenarios. ConvGeN (Schultz et al., 2024) leverages a deep-generative model to learn convex combinations of minority class samples, producing high-quality samples for small tabular datasets while reducing over-generalization compared to SMOTE and GAN-based methods (Good-fellow et al., 2014; Xu et al., 2019). SMOTE-TLNN-DEPSO (Dixit & Mani, 2023) integrates SMOTE with Tomek Links, neural networks, and differential evolution optimization to generate and filter high-quality synthetic samples, enhancing robustness against noisy data. In contrast, un-dersampling methods reduce majority class samples to balance datasets. Spatial Distribution-based Undersampling (SDUS) (Yan et al., 2023) exploits spatial relationships, while Metaheuristic-based Under-Sampling (MHUS) (Soltanzadeh et al., 2023) employs a genetic algorithm to select represen-tative majority samples, minimizing information loss compared to random undersampling (Kubat & Matwin, 1997). Despite their versatility, data-level methods face significant challenges. Over-sampling techniques, such as SMOTE and its variants, may introduce synthetic noise, especially in high-dimensional or overlapping datasets, potentially degrading classifier performance. GAN-based methods often produce inconsistent sample quality, particularly for non-tabular data like images. Undersampling risks discarding critical information, which can be detrimental in datasets with com-plex distributions or extreme imbalance ratios.

**Algorithm-Level Methods** Algorithm-level methods enhance minority class performance by mod-ifying the learning process, offering fine-grained control over model optimization. Cost-sensitive learning assigns higher penalties to misclassifications of minority class samples, prioritizing their correct classification (Elkan, 2001; Zhou & Liu, 2006; Ling & Sheng, 2010). Recent advance-ments address the challenge of specifying fixed cost matrices. Cost-Free Learning (CFL) (Zhang & Hu, 2013) eliminates the need for explicit cost definitions, while Adaptive Threshold Error Costs (ATEC) (Cao et al., 2021) dynamically adjusts error thresholds to adapt to class distributions. Cu-mulative Cost-Sensitive Boosting (AdaCC) (Iosifidis et al., 2022) further advances this paradigm by dynamically adjusting sample weights based on cumulative misclassification costs (false positive and false negative rates), using two variants (AdaCC1 and AdaCC2) to incorporate costs into the weight update formula, achieving superior performance without requiring a predefined cost matrix. Modified loss functions provide an alternative approach. Focal Loss (Lin et al., 2017) reduces the influence of easily classified samples, emphasizing hard-to-classify instances, while Class-Balanced Loss (Cui et al., 2019) reweights losses based on class frequency to balance contributions. Tech-niques such as adaptive learning rates and distribution-aware optimization further refine training parameters to enhance minority class representation (He & Garcia, 2009; Kingma & Ba, 2014; Cao et al., 2019). Algorithm-level methods excel in tailoring model behavior to specific class imbalance scenarios, particularly when domain knowledge is limited. However, their reliance on hyperparam-eter tuning, as seen in Focal Loss and ATEC, can complicate deployment across diverse datasets. Additionally, these methods may struggle to generalize across datasets with extreme imbalance or noisy boundaries, necessitating ensemble strategies to improve robustness.

**Ensemble Learning Methods** Ensemble learning methods combine multiple classifiers to enhance robustness, leveraging diverse models to mitigate class imbalance. Bagging (Breiman, 1996) gener-ates varied subsets via bootstrap sampling, but often underrepresents minority classes in imbalanced datasets, leading to biased models (Sun et al., 2007). Balanced Random Forests (Chen & Breiman, 2004) address this by undersampling the majority class in each subset, improving minority class rep-resentation. Boosting methods, such as AdaBoost (Freund & Schapire, 1997b), iteratively increase weights of hard-to-classify samples, enhancing their classification accuracy. SMOTEBoost (Chawla et al., 2003) integrates SMOTE with Boosting, generating synthetic minority samples in each it-eration to enrich minority class representation. RUSBoost (Seiffert et al., 2010) pairs random un-dersampling with Boosting, offering a computationally efficient alternative. SMOTEWB (Sağlam & Cengiz, 2022) enhances SMOTE with noise detection, dynamically adjusting parameters during Boosting to reduce synthetic noise. MESA (Liu et al., 2020) enhances ensemble learning by em-ploying a meta-sampler to adaptively learn sampling strategies, improving performance on extreme

imbalance scenarios. WSMOTE-ensemble (Abedin et al., 2022) employs Weighted SMOTE with Bagging to generate diverse synthetic samples, improving model diversity. The Contribution-Based Hybrid Resampling Ensemble (CHRE) (Zhao et al., 2025) uses a Globally Unified Data Evaluation (GUDE) algorithm to assess sample contributions based on information and noise levels, guiding oversampling and undersampling within a serial ensemble framework to balance data distribution. Ensemble methods are particularly effective in handling moderately imbalanced datasets by leveraging multiple classifiers to capture diverse patterns. However, they face challenges with extreme class imbalances, where minority class underrepresentation persists, as seen in Bagging and AdaBoost. Methods like SMOTEBoost and CHRE mitigate this by integrating resampling, but their reliance on synthetic samples can introduce noise, particularly in complex or high-dimensional datasets. Furthermore, ensemble methods often incur significant computational costs due to multiple classifier training and hyperparameter tuning, limiting their scalability in large-scale applications.

