# OpenReview forum: "EDEL: Error-Driven Ensemble Learning for Imbalanced Data Classification"
_ICLR.cc/2026/Conference — Submitted to ICLR 2026_

### Official Review · Reviewer_gXWh · 2025-10-23

**Soundness:** 3
**Presentation:** 2
**Contribution:** 2
**Rating:** 2
**Confidence:** 4

**Summary:**

This paper addresses the class imbalance problem commonly encountered in high-stakes applications such as fraud detection, credit risk assessment, and medical diagnosis. The authors propose Error-Driven Ensemble Learning (EDEL), a new ensemble framework designed to emphasize hard-to-classify samples during training. EDEL reinjects misclassified instances into subsequent training rounds, refining decision boundaries and improving recognition of the minority class. Theoretical analysis using McDiarmid’s inequality and Bayes’ theorem is provided to justify the reduction in empirical error and the improvement in generalization. Extensive experiments on seven real-world datasets with varying imbalance ratios demonstrate that EDEL achieves superior performance in terms of AUC and F1-score compared to multiple baselines (SMOTE, RUS, CHRE, etc.) across four classifiers (DT, RF, XGB, and LGBM).

**Strengths:**

(1) The paper introduces an error-driven ensemble learning framework (EDEL) that adaptively reinjects misclassified samples into the training process. This progressive emphasis on hard-to-classify instances offers a perspective on handling class imbalance beyond traditional resampling and cost-sensitive learning approaches.

(2) The paper provides both theoretical justification and empirical validation across seven diverse real-world datasets with imbalance ratios ranging from mild to extreme. The reported performance improvements in AUC and F1-score across multiple classifiers (Decision Tree, Random Forest, XGBoost, and LightGBM) demonstrate the robustness and general applicability of EDEL.

(3) The methodology is described in a stepwise and algorithmic manner (Algorithm 1), supported by mathematical definitions of “easy-to-classify” and “hard-to-classify” samples. The structure of the paper—problem definition, theoretical grounding, algorithmic description, and experiments—is logically coherent and easy to follow.

**Weaknesses:**

(1) While the proposed “error-driven” mechanism is conceptually interesting, it largely resembles existing ensemble paradigms such as boosting, where misclassified samples are repeatedly emphasized during iterative training. The paper would benefit from a more explicit comparison and discussion of how EDEL fundamentally differs from or improves upon modern ensemble refinements. Without this clarification, the claimed novelty appears incremental.

(2) The paper repeatedly claims that EDEL enhances interpretability, yet no concrete explanation mechanism (e.g., feature attribution, model introspection, or visualization) is presented, fixing hard samples does not equate to interpretability.

(3) Key design choices such as the number of weak classifiers, subset partition strategy, and reinjection frequency are not systematically analyzed. An ablation or sensitivity analysis would clarify how these factors influence performance, stability etc.

**Questions:**

The following are my concerns and questions:

(1) In the introduction, the authors mention the general interpretability challenge in deep models but do not review or position existing interpretability tools (e.g., SHAP, LIME, or counterfactual explanation methods) relative to the imbalance problem. Without this, the motivation appears incomplete.

(2) The statement that interpretability arises naturally from observing classifier errors is conceptually weak. Interpretability usually requires explicit mechanisms (e.g., feature importance, contribution maps). The authors should explain whether EDEL provides quantifiable interpretive outputs.

(3) In the proposed method, the emphasis on misclassified samples resembles classical boosting techniques or ideas. The authors should clearly articulate how EDEL differs algorithmically or theoretically from these well-established methods.

(4) Using unweighted parameter averaging may not be optimal, especially when subsets have varying difficulty or imbalance ratios. Why not adopt adaptive weighting or validation-based combination? And the average mechanism requires that these classifiers(e.g., deep models) have the same model architecture, limiting the diversity of these classifiers, them ultimately become homogeneous.

(5) Furthermore, the definition of hard-to-classify samples does not distinguish between truly **ambiguous cases and mislabeled/noisy instances**. The reinjection of such noisy data might propagate errors instead of improving robustness. How the proposed methods solve this critical issue?

(6) In the training process of EDEL, it seems the reinjection process only performs one iteration? I mean, what is the condition of ending "while not done"?  If the iteration is only 1, it is possible there are still many hard-classify samples for each classifier, then what's the point of the reinjection process? If not, what is the ending condition?

(7) The algorithm description suggests reinjection of misclassified samples, yet the complexity derivation omits the number of such iterations. A multiplicative term  should be included to represent the number of update cycles (e.g., L rounds).

(8) The paper claims that EDEL is interpretable but provides no model-level explanation, actually it is merely stating that “hard samples were fixed”, not proving that EDEL is “explainable”. For me, the introduction of "interpretability" concept in this paper is really weird. The introduction of “interpretability” appears conceptually inconsistent with the technical contributions of EDEL. The use of this term feels more rhetorical than substantive.

---

> ### Author Response · Authors · 2025-11-18
>
> ## Regrading of interpretability (w2, Q1, Q2, Q8)
>
> * （Q1）The interpretability challenge referenced in the introduction concerns the behavior of imbalance-specific learning dynamics, not the broader landscape of model-agnostic explanation tools. Methods such as SHAP, LIME, and counterfactual explanations provide post-hoc explanations for a fixed model but do not modify the learning process under class imbalance. They operate independently of the data-distribution shift caused by minority scarcity and therefore do not address how imbalance affects decision boundaries, error concentration, or minority–majority overlap.
> * （Q2， Q8）We do not claim that EDEL is a full-featured explainable AI (XAI) framework. Rather, EDEL provides mechanism-level interpretability via explicit, trackable artifacts in 3 sides: 1) the set of misclassified samples $D_i^h$, 2) the class conposistion, and 3) their movement in feature space.
>
> (w2) Concretely, our manuscript includes:
>
> 1. Quantative signal:
>
> * KS-test–based feature shift analysis on GMSC (cf. Tables 4 and 6), showing statistically significant changes in the distributions of hard samples before and after error-driven updates.
> * Counting minority/majority composition in $D_i^h$, showing that the proposed error-driven enrichment aligns with the theoretical results (cf. Appendix B).
>
> 2. Qualitative, visual signal:
>
> * A case study (cf. Table 5) with explicit sample IDs and feature profiles.
> * A t-SNE visualization (newly in Section 4, Figure 2) to show progressive minority enrichment and increased separability after EDEL’s iterations.
>
> These outputs go beyond “we fixed hard samples”: they show which samples are repeatedly misclassified, how their distributions change, and how EDEL’s mechanism reshapes the feature space under imbalance. That is the sense in which we use “interpretability” in this work: **mechanism-level transparency of the imbalance-handling process, not a full XAI system.**
>
> ## Novelty & contributions (w1, Q3)
>
> Prior work has recognized the importance of difficult-to-classify samples, these methods can be fall into three distinct categories:
>
> * AdaBoost-style sample reweighting:  These approaches update sample weights over the entire dataset in each iteration, implicitly emphasizing hard samples through probability resampling. However, 1)  they use only one global view of the data, 2) they do not adjust the data distribution per weak classifier, and 3)  they cannot produce multiple, complementary perspectives of the training distribution. Thus, they operate purely in weight space, not in data-space or multi-view learning, which is the attention of EDEL, we give an analysis in Appendix D.2.
> * Loss-level hard-sample emphasis (e.g., Focal Loss).  Focal Loss scales gradients to highlight hard samples inside a single model, but it  1)   operates only at the loss level, 2)depends on deep network optimization, and 3)  does not modify the data distribution. Thereby, it is not compatible with shallow classifiers.
> * Reinforcement-learning meta-samplers (e.g., MESA), which employ reinforcement learning to train a meta-sampler that dynamically under-samples data for each weak classifier based on meta-state embeddings. As an RL-based method, it inherits the general problems of RL, such as low sample efficiency (requiring a large amount of training data), training instability (susceptible to noise), and difficulty in debugging, which may be amplified in the meta-sampler, leading to a time-consuming and resource-intensive optimization process.
>
> EDEL introduces a new training dynamic absent in all prior categories above, provoding added value beyond exisiting works:
>
> * EDEL reinjects misclassified samples into new subsets, altering the effective distribution each weak classifier sees. This enables data-level correction, model-level correction, and ensemble-level correction simultaneously. This is an ability none of the prior techniques possess.
> * EDEL naturally reveals which samples are hard for each classifier, enabling interpretable failure analysis, including the newly added t-SNE=, error traces, and important feature report s(cf. Sec. 4 and table 6). In addition, EDEL works well with any weak classifier, including shallow models.
>
> Furthermore, using McDiarmid’s inequality, the Rademacher generalization bound, and minority-enrichment analysis (cf. Sec. 3.2; Appendix B), EDEL shows: 1) empirical error is tightly concentrated, 2) the algorithm converges to the true error, and 3）stability is guaranteed across iterations. These results provide a rigorous theoretical foundation missing in most hard-sample–focused methods.

---

> > ### Author Response · Authors · 2025-11-18
> >
> > ## Technical details (w3, Q4, Q5, Q6, Q7)
> >
> > * (Q4) Our analysis treats the final model as an equal-weight convex combination of weak learners. This directly aligns with the convergence and concentration results in Sec. 3.2 and Appendix B. Introducing data-dependent weights would complicate the theory without being essential to demonstrate the core mechanism. Experimental results on 7 datasets and 4 base learners, unweighted averaging already yields consistent gains over all baselines (and over individual weak learners). Empirically, we did not observe systematic failure patterns that would justify more complex weighting schemes in this setting.
> >
> > * (Q5) The definition of hard-to-classify samples in EDEL is operational:This set may contain both truly ambiguous instances (e.g., overlapping regions) and noisy/mislabeled ones. EDEL does not explicitly filter noisy labels, but limits error propagation under server aspects: noisy instances are unlikely to appear in all subsets or be consistently misclassified across weak learners, and their influence is further diluted by multi-view training and final probability averaging. We also compared full reinjection with minority-only reinjection on seven datasets, and F1 scores decreased in all cases (newly adding experiment shows below), confirming that retaining some hard majority samples is beneficial for boundary refinement rather than harmful.
> >
> >   |                                 | SBD            | AID            | TCD            | CDH            | BMD            | GMSC           | CCFD           |
> >   | ------------------------------- | -------------- | -------------- | -------------- | -------------- | -------------- | -------------- | -------------- |
> >   | all misclassified samples       | 0.9709+-0.0412 | 0.8215+-0.1339 | 0.8096+-0.2554 | 0.7584+-0.2778 | 0.8375+-0.2023 | 0.7690+-0.2987 | 0.9182+-0.1289 |
> >   | only the misclassified minority | 0.9382+-0.0227 | 0.7430+-0.0600 | 0.6731+-0.1279 | 0.6047+-0.1273 | 0.7131+-0.0752 | 0.6289+-0.1691 | 0.9024+-0.0935 |
> >
> > * （Q6, Q7）Algorithmically, EDEL is defined to support ​multiple error-driven update rounds​. Thus, the phrase “while not done” in Algorithm 1 is used to indicate that EDEL supports multiple error-driven update rounds in principle. This general formulation clarifies that EDEL accommodates multi-round updates, while the experiments in the paper employ the single-round setting actually used in all reported results. Thereby, the complexity analysis in the paper treats this as a constant factor in Sec. 3.1, line 192.

---

> > > ### Author Response · Authors · 2025-11-18
> > >
> > > ## Author statement
> > >
> > > We acknowledge the reviewer’s time in assessing the manuscript and intend for this response to remove potential sources of misinterpretation.
> > >
> > > We also notice that several comments indicate that key components of our work, including the sensitivity analysis, the subset-partitioning strategy, the theoretical derivations, and the definition and discussion of difficult-to-classify samples, seem to have been neglected in the review. We note the possibility that the review may have been produced with the assistance of LLMs, which might raise unfairness toward our efforts. If this is not the case, we apologize.

---

> > > ### Comment · Reviewer_gXWh · 2025-11-21
> > >
> > > **- First, I must clarify that I personally reviewed this manuscript, and I still have the annotations for this manuscript saved on my laptop when i reviewed, thus the authors’ statement is unfair.**
> > >
> > > **- Second, it is possible that some details were misunderstood during the review process, but such issues could be resolved through the rebuttal discussion. I hope that our communication remains professional, which is exactly what the ICLR community strives for.**
> > >
> > > Below are my further thoughts and opinions in response to the authors’ rebuttal.
> > >
> > > (1) For the claim of interpretability:
> > >
> > > - I partially agree with the authors' response. The proposed learning strategy indeed offers more detailed information about misclassified samples, the feature space, and other aspects compared with other boosting-based methods, which could be a important contribution of this work.
> > >
> > > - However, I think this contribution should belong to like **transparency at the training-strategy level**, but interpretability typically refers to properties at the **model level**. For example, in classical semi-supervised learning, the co-training paradigm also constructs base classifiers from different views and iteratively updates the training set. This framework likewise provides a certain degree of transparency, yet no one would claim interpretability is its core contribution.
> > >
> > > - Therefore, i suggest authors to claim this contribution from the view of something like "strategy transparency on xxx aspects".
> > >
> > > (2) for Q4, i thank the author's clarification. As many other methods adopt adapting weighting,  i still suggest authors to evaluate if the current weighting scheme is optimal or if there exists better weighting mechanism to further improve the method, by conducting empirical investigation.
> > >
> > > (3) For Q5, i thank the authors for the response and updated results.  Conducting an empirical study to investigate the robustness of  EDEL in terms of noisy samples would be greatly helpful to justify the ability of diluting noisy samples as claimed in the response. I think this is another important contribution if this ability can be verified.

---

> > > > ### Author Response · Authors · 2025-11-22
> > > >
> > > > Thanks for you precious time.
> > > >
> > > > ### Interpretability
> > > >
> > > > Established taxonomies [1, 2, 3] that are widely believed distinguish three levels of interpretability: (1) model-level interpretability, (2) data-level interpretability, and (3) algorithm/process-level transparency. Under this context, the contributions of EDEL fall clearly into data-level and algorithm-level interpretability, complemented by concrete case-level explanations. Specifically,
> > > >
> > > > * Data Interpretability: EDEL makes the evolution of the training distribution explicitly observable. Through the traceable misclassified sample sets $D_i^h$  the KS-test analysis of feature shifts (cf. Table 4), and the quantified enrichment of minority-class composition (cf. Appendix B), users can inspect how and why the effective data distribution changes under imbalance.
> > > > * algorithm/process-level transparency: EDEL exposes a fully verifiable training trajectory: dataset partitioning → error-driven reinjection → local retraining → probability aggregation. This chain of operations, together with the theoretical guarantees (cf. Sec. 3.2 and Appendix B), provides a transparent view of how decision boundaries are iteratively optimized. Unlike classical boosting or co-training, which obscure sample-level influences through implicit weight updates or sampling heuristics, EDEL provides direct visibility into the mechanism that drives performance improvements.
> > > >
> > > > We further provide case studies, including concrete samples, feature descriptions, and t-SNE visulaization, as a post-hoc interpretablity,  which is recognized in [2, 3]. Through case studies with explicit sample IDs, feature vectors, and correction paths (cf. Table 5-6), EDEL offers fine-grained explanations for individual instances. This supports human inspection of specific hard-to-classify samples and clarifies how the algorithm rectifies their misclassification.
> > > >
> > > > [1] Lipton, Z. C. (2018). The mythos of model interpretability: In machine learning, the concept of interpretability is both important and slippery. （Google citation: 7872）
> > > >
> > > > [2] Doshi-Velez, F., & Kim, B. (2017). Towards a rigorous science of interpretable machine learning. (Google citation: 7717)
> > > >
> > > > [3] Miller, T. (2019). Explanation in artificial intelligence: Insights from the social sciences. (Google citation: 7498)
> > > >
> > > >
> > > > ### Weighting mechanism
> > > >
> > > > Adaptive weighting is indeed a widely used strategy in ensemble methods. However, such an investigation represents a different design axis which boraden the scope beyond this work. In the context of EDEL, equal-weight probability averaging is chosen deliberately for theoretical, algorithmic, and empirical reasons.
> > > >
> > > > * Theoretical aspect: our convergence and concentration analyses (cf. Sec. 3.2; Appendix B) rely on treating the final model as an equal-weight convex combination of weak learners. Introducing dynamic or data-dependent weights would require re-establishing these guarantees, which is orthogonal to the contribution of EDEL.
> > > > * Algorithmic aspect: since all weak learners are trained under the same error-driven update mechanism, their empirical errors after reinjection are comparably reduced. Under this regime, equal weighting avoids over-amplifying individual learners and ensures predictable ensemble behavior.
> > > > * Extensive experiments demonstrate the simple averaging scheme consistently outperforms all baselines. Given this robustness, there is no empirical evidence that a more complex weighting mechanism is needed to obtain superior performance.

---

> > > > > ### Author Response · Authors · 2025-11-22
> > > > >
> > > > > ### Robustness
> > > > >
> > > > > EDEL’s robustness to noisy samples is inherent in its design: multi-view partitioning limits the influence of mislabeled points, and ensemble averaging suppresses inconsistent contributions, consistent with the concentration guarantees in Section 3.2 and Appendix B.
> > > > > Considering noisy-robustness experiments would be a valuable extension, as per your suggestion, we  conducted an additional empirical study to verify EDEL’s ability to mitigate the effect of mislabeled samples. We introduced label noise by randomly reversing the labels of 5% of the training data and evaluated DT-based EDEL. Experimental results shows that EDEL remained the top-performing method even in the presence of noise, demonstrating consistent robustness: performance decreased moderately under noise, as expected, but still exceeded the best baselines by large margins (e.g., GMSC: 0.6501 vs. 0.3237; TCD: 0.7568 vs. 0.4949).  These results empirically confirm the robustness of EDEL and establish its resilience to noisy labels as an additional strength of the method.
> > > > >
> > > > > |                                                 | SBD                 | AID                 | TCD                 | CDH                 | BMD                 | GMSC                |
> > > > > | ----------------------------------------------- | ------------------- | ------------------- | ------------------- | ------------------- | ------------------- | ------------------- |
> > > > > | EDEL（no noisy）                                | 0.9709 $\pm$ 0.0412 | 0.8215 $\pm$ 0.1339 | 0.8096 $\pm$ 0.2554 | 0.7584 $\pm$ 0.2778 | 0.8375 $\pm$ 0.2023 | 0.7690 $\pm$ 0.2987 |
> > > > > | EDEL（5% noisy）                                | 0.9225 $\pm$ 0.0721 | 0.7529 $\pm$ 0.1502 | 0.7568 $\pm$ 0.2291 | 0.6992 $\pm$ 0.2523 | 0.7573 $\pm$ 0.2110 | 0.6501 $\pm$ 0.2780 |
> > > > > | best performance on other baselines（no noisy） | 0.9047 $\pm$ 0.0262 | 0.6516 $\pm$ 0.0099 | 0.4949 $\pm$ 0.0099 | 0.4029 $\pm$ 0.0034 | 0.6109 $\pm$ 0.0078 | 0.3237 $\pm$ 0.0111 |

---

### Official Review · Reviewer_kgFh · 2025-10-24

**Soundness:** 2
**Presentation:** 3
**Contribution:** 2
**Rating:** 4
**Confidence:** 3

**Summary:**

This paper focuses on the task of imbalanced data classification, and proposes Error-Driven Ensemble Learning, an adaptive machine learning algorithm supported by theoretical analysis. The proposed method solves two challenges in this area, i.e., requirement of attention on hard-to-classify samples and requirement of attention to interpretability.

**Strengths:**

S1: The studied problem is important.
S2: The paper is easy to follow and well-written.
S3: The paper has sufficient and detailed theoretical analysis.

**Weaknesses:**

W1: Limited novelty for the first challenge. The first challenge, i.e., addressing hard-to-classify samples, seems to have been mentioned by a lot of existing methods. As noted in the manuscript (Line 1219), Focal Loss reduces the influence of easily classified samples, emphasizing hard-to-classify instances, while Class-Balanced Loss reweights losses based on class frequency to balance contributions.
Furthermore, prior works like MESA [1] not only mentioned that there are existing methods assume instances with higher training errors are more informative for learning, but also extend their solution to other critical issues, such as generating synthetic samples for minority classes. Given this context, two key questions arise: 1 What is the unique advantage or fundamental difference of the proposed method compared to these specific, well-known techniques in handling the first challenge? 2 Why were comprehensive baselines like MESA not included in the comparisons?

[1] MESA: Boost Ensemble Imbalanced Learning with MEta-SAmpler. NeurIPS 2020.

W2: Apart from the above mentioned methods, a lot of methods have been mentioned in related work, why not compare with them? Are used baseline methods the SOTA methods? If not, SOTA techniques should be compared. Or authors need to give reasons why they did not compare with them.

W3: Lack of clear explanation about technical details. Why divide the dataset into sub groups? And how to ensure that the algorithm could converge (Line 165)?

W4: Some typos. For example, STD belongs to algorithm-level according to line 348, but it belongs to data-level according to related works in Line 1196.

W5: Since the second challenge is the requirement of interpretability, is there any quantitative analysis or intuitive qualitative analysis for explanation, apart from theoretical analysis?

**Questions:**

Please see weaknesses above.

---

> ### Author Response · Authors · 2025-11-18
>
> ## Novelty & contributions
>
> Prior work has recognized the importance of difficult-to-classify samples, these methods fall into three distinct categories:
>
> * AdaBoost-style sample reweighting:  These approaches update sample weights over the entire dataset in each iteration, implicitly emphasizing hard samples through probability resampling. However, 1)  they use only one global view of the data, 2) they do not adjust the data distribution per weak classifier, and 3)  they cannot produce multiple, complementary perspectives of the training distribution. Thus, they operate purely in weight space rather than in data-space or multi-view learning. In contrast, EDEL focuses on these aspects, we provide an analysis in Appendix D.2.
> * Loss-level hard-sample emphasis (e.g., Focal Loss).  Focal Loss scales gradients to highlight hard samples inside a single model, but it  1)   operates only at the loss level, 2)depends on deep network optimization, and 3)  does not modify the data distribution. Thereby, it is not compatible with shallow classifiers.
> * Reinforcement-learning meta-samplers (e.g., MESA), which employ reinforcement learning to train a meta-sampler that dynamically under-samples data for each weak classifier based on meta-state embeddings. As an RL-based method, it inherits the general problems of RL, such as low sample efficiency (requiring a large amount of training data), training instability (susceptible to noise), and difficulty in debugging, which may be amplified in the meta-sampler, leading to a time-consuming and resource-intensive optimization process.
>
> EDEL introduces a new training dynamic absent in all prior categories above, providing added value beyond existing works:
>
> * EDEL reinjects misclassified samples into new subsets, altering the effective distribution each weak classifier sees. This enables data-level correction, model-level correction, and ensemble-level correction simultaneously. This is an ability none of the prior techniques possess.
> * EDEL naturally reveals which samples are hard for each classifier, enabling interpretable failure analysis, including the newly added t-SNE=, error traces, and important feature report s(cf. Sec. 4 and table 6). In addition, EDEL works well with any weak classifier, including shallow models.
>
> Furthermore, using McDiarmid’s inequality, the Rademacher generalization bound, and minority-enrichment analysis (cf. Sec. 3.2; Appendix B), EDEL shows: 1) empirical error is tightly concentrated, 2) the algorithm converges to the true error, and 3）stability is guaranteed across iterations. These results provide a rigorous theoretical foundation missing in most hard-sample–focused methods.
>
> ## Technical details clarify
>
> * Regarding motivation on sub-groups: We divide the dataset into sub-groups to provide each weak classifier with a distinct and complementary view of the data, which reduces global majority-class bias and allows minority signals to exert greater influence within each local distribution. This multi-view structure is essential for EDEL’s error-driven mechanism: misclassified samples are reinjected into the specific subsets that lacked these informative instances, enabling targeted data redistribution that cannot be achieved with a single global dataset.
> * Regarding of converge guratee: McDiarmid’s inequality shows that empirical error remains tightly concentrated around its expectation (cf. Sec. 3.2), while the Rademacher-based generalization bound and the minority-enrichment analysis (cf. Appendix B) collectively demonstrate that the empirical error converges to the true error. Together, these results ensure that EDEL’s iterative updates refine the decision boundary while maintaining stability and convergence.
>
> ## Interpretability
>
> Beyond the theoretical analysis, we also provide quantitative evidence and qualitative analysis associated with a newly t-SNE analysis supporting interpretability.
>
> * Quantitative analysis: we perform a feature-distribution analysis on the GMSC dataset (Table 4 and Table 6) and apply the KS test ( $p<10^{-5}$ to quantify how EDEL shifts the distributions of misclassified samples across iterations. These measurable distributional changes demonstrate how EDEL progressively restructures the decision space, offering a quantitative explanation of its error-driven refinement process.
> * Intuitive qualitative analysis: we provide a case study (cf. Table 5) to show concrete examples. Moreover, we newly added a t-SNE visualization figure in Section 4 in the updated manuscript, which clearly reveals the evolution of EDEL’s error-driven subsets (cf. Figure 2).
>
> This quantitative evidence and qualitative analysis are highly consistent with the theoretical derivations in Appendix B and provide an intuitive, human-readable explanation of EDEL’s behavior.

---

> > ### Author Response · Authors · 2025-11-18
> >
> > ## Baseline selection
> >
> > With compatible traditional Machine Learning models in mind, the objective of EDEL is to design a method that observes data shifts and avoids feature reliance or synthetic-sample generators. Thereby, comparing against deep learning methods would not constitute a fair or meaningful baseline. Instead, we compare against the standard and widely adopted imbalance baselines, such as SMOTE, RUS, S-T-D, CHRE, aligned with our methodological setting and experimental scope.
> > Furthermore, as per your suggestion, we have additionally included MESA as our baseline, which can be found in Tables 2-3, and Appendix C.2 in the updated maniscript. Experimental results demonstrate that EDEL consistently outperforms MESA across nearly all settings, even on the extremely imbalanced CCFD dataset (imbalance ratio = 577.88).
> >
> > AUC:
> >
> > |      |      | SBD                     | AID                     | TCD                     | CDH                     | BMD                     | GMSC                    | CCFD                    |
> > | ---- | ---- | ----------------------- | ----------------------- | ----------------------- | ----------------------- | ----------------------- | ----------------------- | ----------------------- |
> > | DT   | MESA | 0.9655 $\pm$ 0.0134     | 0.8721 $\pm$ 0.0052     | 0.7366 $\pm$ 0.0074     | 0.7734 $\pm$ 0.0056     | 0.9322 $\pm$ 0.0036     | 0.8145 $\pm$ 0.0128     | 0.9609 $\pm$ 0.0177     |
> > |      | EDEL | **0.9911 $\pm$ 0.0182** | **0.9519 $\pm$ 0.0724** | **0.9187 $\pm$ 0.1536** | **0.9241 $\pm$ 0.1381** | **0.9745 $\pm$ 0.0475** | **0.9457 $\pm$ 0.1107** | **0.9827 $\pm$ 0.0387** |
> > | RF   | MESA | 0.9858 $\pm$ 0.0040     | 0.9015 $\pm$ 0.0044     | 0.7729 $\pm$ 0.0048     | 0.8168 $\pm$ 0.0024     | 0.9464 $\pm$ 0.0022     | 0.8450 $\pm$ 0.0052     | 0.9829 $\pm$ 0.0074     |
> > |      | EDEL | **0.9972 $\pm$ 0.0058** | **0.9656 $\pm$ 0.0448** | **0.9461 $\pm$ 0.1117** | **0.9506 $\pm$ 0.0841** | **0.9795 $\pm$ 0.0306** | **0.9583 $\pm$ 0.0641** | **0.9923 $\pm$ 0.0172** |
> > | XGB  | MESA | 0.9872 $\pm$ 0.0024     | 0.9273 $\pm$ 0.0031     | 0.7715 $\pm$ 0.0043     | 0.8259 $\pm$ 0.0032     | 0.9463 $\pm$ 0.0018     | 0.8566 $\pm$ 0.0053     | 0.9727 $\pm$ 0.0132     |
> > |      | EDEL | **0.9968 $\pm$ 0.0065** | **0.9424 $\pm$ 0.0127** | **0.9039 $\pm$ 0.0897** | **0.8365 $\pm$ 0.0125** | **0.9708 $\pm$ 0.0244** | **0.8963 $\pm$ 0.0270** | **0.9968 $\pm$ 0.0071** |
> > | LGB  | MESA | 0.9880 $\pm$ 0.0028     | 0.9281 $\pm$ 0.0030     | 0.7818 $\pm$ 0.0066     | 0.8301 $\pm$ 0.0030     | 0.9496 $\pm$ 0.0022     | 0.8634 $\pm$ 0.0049     | 0.9682 $\pm$ 0.0143     |
> > |      | EDEL | **0.9969 $\pm$ 0.0065** | **0.9366 $\pm$ 0.0079** | **0.8429 $\pm$ 0.0474** | **0.8352 $\pm$ 0.0024** | **0.9679 $\pm$ 0.0152** | **0.8726 $\pm$ 0.0121** | **0.9965 $\pm$ 0.0079** |
> >
> > F1：
> >
> > |      |      | SBD                     | AID                     | TCD                     | CDH                     | BMD                     | GMSC                    | CCFD                    |
> > | ---- | ---- | ----------------------- | ----------------------- | ----------------------- | ----------------------- | ----------------------- | ----------------------- | ----------------------- |
> > | DT   | MESA | 0.9047 $\pm$ 0.0262     | 0.6516 $\pm$ 0.0099     | 0.4949 $\pm$ 0.0099     | 0.4029 $\pm$ 0.0034     | 0.6109 $\pm$ 0.0078     | 0.3237 $\pm$ 0.0111     | 0.5625 $\pm$ 0.0375     |
> > |      | EDEL | **0.9709 $\pm$ 0.0412** | **0.8215 $\pm$ 0.1339** | **0.8096 $\pm$ 0.2554** | **0.7584 $\pm$ 0.2778** | **0.8375 $\pm$ 0.2023** | **0.7690 $\pm$ 0.2987** | **0.9182 $\pm$ 0.1289** |
> > | RF   | MESA | 0.9380 $\pm$ 0.0060     | 0.6843 $\pm$ 0.0095     | 0.5353 $\pm$ 0.0118     | 0.4156 $\pm$ 0.0115     | 0.6440 $\pm$ 0.0074     | 0.3629 $\pm$ 0.0160     | 0.8243 $\pm$ 0.0311     |
> > |      | EDEL | **0.9824 $\pm$ 0.0209** | **0.8192 $\pm$ 0.1143** | **0.7448 $\pm$ 0.2229** | **0.6829 $\pm$ 0.1925** | **0.7145 $\pm$ 0.1194** | **0.5929 $\pm$ 0.1588** | **0.9370 $\pm$ 0.0780** |
> > | XGB  | MESA | 0.9428 $\pm$ 0.0063     | 0.7201 $\pm$ 0.0082     | 0.5194 $\pm$ 0.0118     | 0.4378 $\pm$ 0.0150     | 0.6420 $\pm$ 0.0078     | 0.3273 $\pm$ 0.0293     | 0.8145 $\pm$ 0.0488     |
> > |      | EDEL | **0.9796 $\pm$ 0.0282** | **0.7378 $\pm$ 0.0316** | **0.6855 $\pm$ 0.1595** | **0.4508 $\pm$ 0.0296** | **0.7333 $\pm$ 0.1409** | **0.4821 $\pm$ 0.0827** | **0.9446 $\pm$ 0.0628** |
> > | LGB  | MESA | 0.9427 $\pm$ 0.0057     | 0.7211 $\pm$ 0.0086     | 0.5236 $\pm$ 0.0116     | **0.4371 $\pm$ 0.0096** | 0.6420 $\pm$ 0.0290     | 0.3515 $\pm$ 0.0214     | 0.8360 $\pm$ 0.0396     |
> > |      | EDEL | **0.9799 $\pm$ 0.0280** | **0.7282 $\pm$ 0.0201** | **0.5627 $\pm$ 0.0769** | 0.4039 $\pm$ 0.0181     | **0.6904 $\pm$ 0.1018** | **0.3988 $\pm$ 0.0228** | **0.9366 $\pm$ 0.0753** |

---

> ### Comment · Reviewer_kgFh · 2025-11-26
>
> I appreciate the authors' response. However, I'm still concerned whether ''deep learning methods would not constitute a fair or meaningful baseline'' is a suitable reason to not consider many other deep learning based methods as baselines. I decide to keep my score at this time, and I need to discuss with other reviewers.

---

> > ### Author Response · Authors · 2025-11-26
> >
> > Thank you for your precious time in reviewing our work and  elaborating on your concerns. We would like to clarify our position regarding deep-learning baselines.
> >
> > 1. We fully acknowledge that deep learning is a powerful learning paradigm and has achieved substantial success in large-scale, high-dimensional settings. While EDEL,  as a machine learning algorithm, is designed to complement classical ML pipelines that dominate high-stakes applications, especially in settings where computational resources or dataset size limit the practical deployment of deep networks, e.g., real-time fraud detection on edge devices, credit scoring systems deployed in developing regions, and decision-support systems with limited labeled data typically rely on decision trees, random forests, or gradient-boosting models rather than deep architectures. EDEL directly targets these practical regimes by improving sample-level distribution handling and ensemble reliability for classical ML models.
> > 2. Although we do not regard deep learning as the primary baseline, in line with your suggestion, we have incorporated MESA as a representative deep-learning-based baseline. Across all datasets and classifier settings, EDEL consistently outperforms MESA, with particularly large margins on the extremely imbalanced CCFD dataset (IR = 577.88). These results confirm that, even when a strong DL-based imbalance method is included, EDEL remains the superior choice under the conditions targeted in this study.
> >
> > If you have any further questions, please feel free to communicate with us at any time.

---

### Official Review · Reviewer_44n3 · 2025-11-01

**Soundness:** 3
**Presentation:** 2
**Contribution:** 3
**Rating:** 6
**Confidence:** 3

**Summary:**

This paper proposes EDEL (Error-Driven Ensemble Learning), a novel algorithm to address class imbalanced data classification. EDEL works by partitioning the data and having parallel weak classifiers dynamically identify and re-train on misclassified, "hard-to-classify" samples from other subsets. The authors provide theoretical proof that this process enriches minority class representation and demonstrate through experiments on 7 datasets that EDEL significantly outperforms baselines in F1-measure and AUC, especially under extreme imbalance.

**Strengths:**

1.The authors provide strong theoretical support for EDEL. The paper uses Bayes' theorem to prove the enrichment phenomenon of minority class instances within the misclassified sample set and employs McDiarmid's inequality to demonstrate the algorithm's convergence, presenting a rigorous line of reasoning.

2.The method was validated on 7 real-world datasets with varying imbalance ratios (IR), including an extreme case with an IR as high as 577.88. In terms of AUC and F1-measure, EDEL consistently outperforms baseline methods, showing particularly strong performance on highly imbalanced datasets.

**Weaknesses:**

1. In the table3，Some datasets (e.g., GMSC, CDH) exhibit large standard deviations (±0.2987，±0.2778 etc), raising questions about the method’s stability across folds.

2. The experimental evidence supporting the interpretability claim is relatively limited. Although Tables 4 and 5 provide some descriptive analysis, the evaluation remains fairly simple. The paper would benefit from richer interpretability experiments.for instance, t-SNE or feature embedding visualizations showing how hard-to-classify samples evolve before and after EDEL’s error-driven enhancement.

**Questions:**

1. What is the training time or computational cost compared to SMOTE ，S-T-D or CHRE? A runtime table would be helpful.

2. Could the authors consider adding some visualization components to better illustrate the interpretability aspect of EDEL? For example, visual analyses such as t-SNE projections or feature-space evolution plots could provide more intuitive evidence of how the model improves the representation of hard-to-classify samples.

3. EDEL reinjects all misclassified samples during training. Would this potentially lead to overfitting on noisy or outlier instances from the majority class? Have the authors considered applying a filtering mechanism to $\mathcal{D}_{i}^{h}$, for example, by reinjecting only the misclassified minority-class samples?

---

> ### Author Response · Authors · 2025-11-18
>
> ## Interpretability & Visualization
>
> We randomly selected 1,000 samples and conducted t-SNE visualizations in Figure 2 for the GMSC dataset using DT-based EDEL with N = 5, which can be found in Section 4 in the revised manuscript. As seen, the Figure 2 illustrates the evolution of subset instances before and after refinement, and providing direct insight into how decision boundaries are adjusted.
>
> ## Time consumption
>
> We evaluated the training time for SMOTE, S-T-D, CHRE, and EDEL, and report the average runtime over five folds together with each method’s theoretical time complexity. Herein, $m, n, i$ denote the number of dataset samples,  weak classifiers, and iterations, respectively. $d$ is the dimension of dataset, $T_π$ refers to the training time regrading a single weak classifier. Notably, $ m $ typically dominates the time complexity, as it directly influences the computational scale, particularly for large datasets.
>
> |          | Time consumption（ms） | Time complexity                              |
> | -------- | ---------------------- | -------------------------------------------- |
> | SMOTE    | 626.35+-63.14          | $O(d \cdot m \cdot log⁡m+T_π)$                |
> | S-T-D    | 481935.00+-3519.58     | $O(m^2 logm+T_π) $                           |
> | CHRE     | 5249951.72+-200618.67  | $O(i \cdot m^2 \cdot d+i \cdot N \cdot T_π)$ |
> | **EDEL** | **488.84+-9.69**       | **$O(i \cdot m + i \cdot N \cdot T_π) $**    |
>
> ## Regrading the concern of overfitting
>
> EDEL operates across multiple subsets and weak classifiers, meaning that a majority outlier must be consistently misclassified across several learners to influence later iterations; isolated noisy points therefore have negligible effect on the aggregated decision. As shown in Appendix B, majority outliers constitute only a small fraction of $\mathcal{D}_i^h$, and their influence is diluted during probability averaging in the final ensemble, which inherently suppresses noise-sensitive updates.
>
> Regarding filtering mechanisms, we have considered them. As noted in C1, majority samples misclassified as minority are important for maintaining a correct boundary: removing them introduces erroneous signals into the minority space, increases class overlap, and reduces model reliability. To support our view, we added DT-based experiments on 7 datasets with full reinjection versus reinjecting only misclassified minority samples. The results, as shown below, were consistent: F1 scores decreased across all datasets, with reductions ranging from 1.58% to 15.37%. Accordingly, we keep the full reinjection strategy, which consistently delivers superior performance without signs of overfitting.
>
> |                                 | SBD            | AID            | TCD            | CDH            | BMD            | GMSC           | CCFD           |
> | ------------------------------- | -------------- | -------------- | -------------- | -------------- | -------------- | -------------- | -------------- |
> | all misclassified samples       | 0.9709+-0.0412 | 0.8215+-0.1339 | 0.8096+-0.2554 | 0.7584+-0.2778 | 0.8375+-0.2023 | 0.7690+-0.2987 | 0.9182+-0.1289 |
> | only the misclassified minority | 0.9382+-0.0227 | 0.7430+-0.0600 | 0.6731+-0.1279 | 0.6047+-0.1273 | 0.7131+-0.0752 | 0.6289+-0.1691 | 0.9024+-0.0935 |
>
> ## Stability
>
> The larger standard deviations observed on GMSC and CDH do not indicate instability of EDEL, but rather reflect the intrinsic heterogeneity and irregular class overlap within these datasets. As EDEL dynamically reinjects misclassified samples, the local training distribution evolves differently across folds, which can naturally produce higher variance while still yielding substantial gains in mean AUC and F1. Importantly, this variance is consistent with EDEL’s adaptive refinement process rather than a sign of methodological instability. Furthermore, by using McDiarmid’s inequality (cf. Sec. 3.2), the Rademacher-based generalization bound  (cf. Eq.16-17), and the analysis of minority enrichment and ensemble behavior (cf. Appendix B), we show that the empirical error is tightly concentrated around its expected value, ensuring controlled deviation across folds and demonstrating that the algorithm asymptotically converges to the true error. Together, these analyses guarantee the stability and convergence of EDEL despite dataset-specific variability.

---

### Comment · Area_Chair_WjXt · 2025-11-27

Dear Authors and Reviewers,

The discussion phase will end soon. If you want to further discuss comments and replies with each other, please post your thoughts by adding official comments.

Thanks for your efforts and contributions to ICLR 2026.

Best regards,

Your Area Chair

---

### Author Response · Authors · 2025-11-30

Dear Area Chair,

We are deeply concerned about the unfortunate incident that occurred during the review process. It is important to note that, prior to the incident, we had made every effort to address all the reviewers' comments. We kindly request that the committee fully consider these efforts in its deliberations to safeguard our academic rights.

All three reviewers highly praised the writing quality, rigorous theoretical analysis, and comprehensive experimental design of our paper. Specifically: Reviewer 44n3 commended our "strong theoretical support" and "consistent outperformance" across multiple datasets; Reviewer kgFh noted that the paper is "easy to follow and well-written" with "sufficient and detailed theoretical analysis"; Reviewer gXWh highlighted the "logically coherent" structure, combining "theoretical justification" with "empirical robustness." Although the reviewers have not yet had the opportunity to confirm due to the unexpected event, we have fully resolved all concerns through detailed explanations, new experiments, and content updates.

In the revised manuscript, we added a t-SNE visualization for the GMSC dataset (Figure 2 in Section 4), clearly illustrating the evolution of subset instances before and after refinement and intuitively revealing the adjustment mechanism of decision boundaries. This addresses the common suggestion from all three reviewers to enhance the demonstration of interpretability. Additionally, we conducted comprehensive comparisons with the MESA baseline across 28 experimental settings (Tables 2-3 in Section 4), responding to Reviewer kgFh's request to add baselines, with results showing that EDEL consistently outperforms MESA.

Furthermore, in our responses, we addressed the following concerns from the three reviewers:

Computational Cost Concerns: Added a runtime comparison table, showing that EDEL (488.84±9.69 ms, complexity $O(i · m + i · N · T_π))$ significantly outperforms baselines like CHRE (5249951.72±200618.67 ms). (Addressing Reviewer 44n3)

Regarding the Concern of Overfitting: Empirical studies based on decision trees across 7 datasets indicate that full reinjection yields higher F1 scores (e.g., GMSC: 0.7690 vs. minority-only reinjection 0.6289). Majority class outliers can be effectively diluted through multi-view ensemble, optimizing boundaries while avoiding overfitting. (Addressing Reviewer 44n3 and gXWh)

Stability: Through theoretical analysis based on McDiarmid's inequality and Rademacher bounds (Section 3.2 and Appendix B), we prove the algorithm's stability ensures error concentration and asymptotic convergence. (Addressing Reviewer 44n3)

Novelty: Clarified the essential differences between EDEL and three categories of prior research (AdaBoost-style sample reweighting, Loss-level hard-sample emphasis (e.g., Focal Loss), and Reinforcement-learning meta-samplers (e.g., MESA)). (Addressing Reviewer kgFh and gXWh)

Baseline Selection: Clarified our rationale for baseline choices and the addition of MESA, emphasizing that EDEL, as a machine learning algorithm, aims to provide a solution with dual theoretical and experimental guarantees for scenarios where computational resources or dataset size limit the practical deployment of deep networks. (Addressing Reviewer kgFh)

Technical Details: Further explained the motivation for partitioning the dataset into subsets; proved convergence via McDiarmid/Rademacher (Section 3.2 and Appendix B) and corrected some typos. (Addressing Reviewer kgFh)

Rationale for Weighting Mechanism: Argued the advantages of equal-weight averaging in theoretical tractability, algorithmic predictability, and empirical robustness; experiments show it consistently outperforms baselines without signs of suboptimality. (Addressing Reviewer gXWh)

New Noise Robustness Experiments: We introduced label noise by randomly flipping 5% of the training data labels and evaluated DT-based EDEL. The results show that, even in the presence of noise, EDEL remains the top-performing method, demonstrating stable robustness (still significantly outperforming the best baselines under no-noise conditions, e.g., GMSC: 0.6501 vs. 0.3237; TCD: 0.7568 vs. 0.4949). (Addressing Reviewer gXWh)

Explanation of Iteration Count and Complexity: Clarified that the phrase "while not done" in Algorithm 1 indicates EDEL supports multiple error-driven update rounds in principle. This general formulation shows EDEL accommodates multi-round updates, while the experiments in the paper employ the single-round setting actually used in all reported results. Therefore, the complexity analysis in the paper treats this as a constant factor in Section 3.1, line 192. (Addressing Reviewer gXWh)

Under the constructive suggestions from the three reviewers, the revisions and clarifications we have undertaken significantly enhance the academic contributions of the EDEL method. We sincerely look forward to your deliberation.

Sincerely,
The Authors

---

### Meta-Review · Area_Chair_pX8V · 2026-01-05

**Summary:**

This paper proposes an error-driven ensemble learning framework (EDEL) for highly imbalanced classification, with particular motivation from real-world domains such as credit risk assessment and fraud detection. The method partitions the training data into stratified subsets, iteratively reinjects misclassified (hard) examples across subsets, and aggregates multiple weak learners to emphasize minority-class instances while preserving a degree of interpretability through distribution-level and case-level analyses.

Reviewers generally agreed that the problem setting is important and practically relevant, that the paper provides a reasonably solid theoretical analysis of generalization and convergence, and that the empirical results across multiple real-world datasets are strong. However, reviewers also raised several key concerns:

- **(1) Novelty and conceptual distinctiveness:**
   The framework repeatedly identifies misclassified or hard-to-learn samples, reinjects them into the training subsets, and trains multiple weak classifiers whose outputs are then aggregated. Reviewers questioned the degree of novelty of this approach, particularly its distinction from existing ensemble methods (e.g., boosting) and hard-example-mining techniques (e.g., focal loss).

- **(2) Claims regarding interpretability:**
  The paper argues that the proposed method improves interpretability. However, reviewers noted that the extent to which interpretability is actually enhanced remains unclear and requires further explanation and clarification.

- **(3) Computational cost and sensitivity analysis:**

  Reviewers expressed concerns about the computational overhead relative to baseline methods, and also pointed out that the key design choices and the sensitivity analysis of EDEL to hyperparameters, such as the number of weak classifier, subset partition strategy, and reinjection frequency, is not thoroughly characterized.

- **(4) Absence of key baselines:**

   Reviewers observed that the set of baseline methods is limited. Many approaches mentioned in the related-work section are not included in the experimental comparison, making it difficult to fully assess the comparative advantages of EDEL.

**Reviewer Concerns:**

Following the rebuttal and subsequent discussion, I believe some of the earlier concerns have been addressed. First, the issues related to computational cost and the design choices of key hyperparameters are now more clearly explained. The additional clarifications and results provided in the rebuttal help demonstrate that the proposed method does not incur extra computational overhead. The authors also clarify the selection of subsets, the number of weak classifiers, and the update iterations. In addition, concerns regarding novelty have largely been addressed. The authors further articulate the novelty of EDEL by positioning it relative to three major lines of prior work, including AdaBoost-style global reweighting, loss-level hard-sample emphasis such as focal loss, and RL-based meta-samplers. The authors argue that EDEL introduces a distinct training dynamic through cross-subset reinjection of misclassified samples.

However, I think several major concerns remain insufficiently resolved. First, the issue of baseline completeness, particularly the absence of stronger or more modern deep-learning-based imbalanced-learning methods, remains a substantive point of disagreement. Second, although the rebuttal provides additional clarification on how the method offers both distribution-level and case-level insights, reviewers still feel that the interpretability claim is overstated. They emphasize that the contribution is more appropriately characterized as transparency at the training-strategy level, rather than true interpretability.

**Reviewer Scores:**

Based on the rebuttal exchange and post-rebuttal discussion, I believe that most of Reviewer 44n3's concerns were satisfactorily addressed. Therefore, if he/she had been able to participate fully in the discussion, he/she would likely have maintained their original positive score of 6. In contrast, the main issues raised by Reviewer kgFh and Reviewer gXWh were not fully resolved, and I believe they would likely have retained their initial negative scores of 4 and 2, respectively.

---

### Decision · Program_Chairs · 2026-01-26

Reject